# The influence of dune lee side shape on time-averaged velocities and turbulence

Alice Lefebvre [1], Julia Cisneros [2]

[1]MARUM – Center for Marine Environmental Sciences, University of Bremen, Bremen, Germany
[2]Department of Geological Sciences, Jackson School of Geosciences, UT Austin, Austin, TX 78712, USA

*Correspondence to*: Alice Lefebvre (alefebvre@marum.de)

**Abstract.** Underwater dunes are found in various environments with strong hydrodynamics and movable sediment such as rivers, estuaries and continental shelves. They have a diversity of morphology, ranging from low to high-angle lee sides, and sharp or rounded crests. Here, we investigate the influence of lee side morphology on flow properties (time-averaged velocities

and turbulence). To do so, we carried out a large number of numerical simulations of flow over dunes with a variety of morphologies using Delft3D. Our results show that the value of the mean lee side angle, as well as the value and position of the maximum lee side angle, have an influence on the flow properties investigated. We propose a classification with 3 types of dunes: (1) low-angle dunes (mean lee side < 10°), over which there is generally no flow separation and over which only little turbulence is created; (2) intermediate-angle dunes (mean lee side 10-17°) over which an intermittent flow separation is

likely over the trough; and (3) high-angle dunes (mean lee side > 17°) over which the flow separates at the brink point and reattaches shortly after the trough, and over which turbulence is high. The influence of maximum lee side slope value and position on flow characteristics depends on dune type. We discuss the implications of the proposed dune classification on the interaction between dune morphology and flow.

## 1 Introduction

Underwater dunes are observed in many shallow and deep-water environments, such as rivers, estuaries, tidal inlets, continental shelves and slopes. They form in systems with beds composed of sediment ranging from coarse silt to fine gravel where the hydrodynamics are strong enough to produce sufficient bedload transport. Until recently, most studies focussed on so-called "high-angle dunes" which possess a steep lee side with slopes of around 30° (angle-of-repose) over which sediment is avalanching (Kleinhans, 2004). These dunes commonly form in small rivers and in flumes (Naqshband et al., 2014a; Van Der

Mark et al., 2008). Over such dunes, flow separation and recirculation over the lee side produces a turbulent wake and induces bedform roughness. However, many dunes have recently been observed to be "low-angle dunes" with lee side angles much lower than the angle-of-repose. Over low-angle dunes, flow separation is absent or intermittent, and turbulence and roughness are lower than over high-angle dunes (Kwoll et al., 2016; Lefebvre and Winter, 2016). In large rivers, mean lee side angles are commonly between 5 and 20° (Cisneros et al., 2020). This is illustrated in Figure 1 by data from the Mississippi and Waal

Rivers which had the steepest and gentlest mean lee side slopes of the six rivers investigated by Cisneros et al. (2020). Bedform lee sides in tidal environments are also often low, with typical values of 2 to 20° (e.g. Lefebvre et al., 2021, Figure 1c;

Dalrymple and Rhodes, 1995; Franzetti et al., 2013; Damen et al., 2018). In addition, contrary to previous simplifications of lee side shape, the lee side is rarely a straight line but rather made of several steeply and gently sloping portions (Lefebvre et al., 2016). Typically, a comparatively steep slope is observed somewhere along the lee side with gentler slopes towards the

crest and/or the trough. The steep slope is often referred to as the "slip face" as sediment is thought to avalanche or slip over this part of the lee side. Although this is true for slope close to the angle-of-repose (Kleinhans, 2004), the transport mechanisms over gentler slopes are still under discussion (Kostaschuk and Venditti, 2019). Therefore, we refer to this slope as the "steep portion" in an effort to not make any assumption on the type of sediment transport along the dune lee side (see also Lefebvre et al., 2021). The maximum lee side angle value and position can be used to characterise the steep portion characteristics. In

large rivers, the maximum lee side angle is generally less than the angle-of-repose (e.g. Figure 1a and b), with an average maximum lee side angle calculated by Cisneros et al. (2020) of 20.5°. In constrained tidal environments (e.g. estuaries and tidal inlets), maximum lee side angles are usually less than 20° (Lefebvre et al., 2021; Prokocki et al., 2022; Dalrymple and Rhodes, 1995) (Figure 1c). In marine environments, there has not yet been a systematic report of mean or maximum angles, in part due to the lack of consistent high-resolution data which may be used to precisely calculate the maximum slope. The

mean and maximum lee side angles will vary depending on dune asymmetry, the absolute and relative strength of ebb and flood currents, the influence of other currents (e.g. river flows, wave-related currents), sediment size and tidal phase. In any case, it should be noted that in tidal environments, flow reverses from one tidal phase to the next. However, large dunes usually stay oriented in one direction during the whole tidal cycle. Therefore, in case of asymmetric bedforms, lee side may be steep during one tidal phase and gentle during the following tidal phase. As a result, a wide range of mean and maximum lee side

angles are likely to be found in marine environments.

Interestingly, the shapes of river (Figure 1d) and estuarine dunes (Figure 1e) differ. River dunes have their steepest slope close to the trough and a rounded crest (Cisneros et al., 2020), whereas estuarine dunes have their steepest slope close to the crest and a sharp crest (Dalrymple and Rhodes, 1995; Lefebvre et al., 2021; Prokocki et al., 2022). Lefebvre et al. (2021) suggested that this is created by the difference in sediment transport direction (Figure 1): in rivers, sediment is systematically eroded

from the crest and deposited over the lee side by the unidirectional currents, creating a rounded crest; in estuaries and other tidal environments, sediment at the crest is being mobilised at each tidal phase but only for a short time in each direction resulting in the sharp crest morphology. In open marine environments, various morphologies are found, from sharp to round crests (Van Landeghem et al., 2009; Zhang et al., 2019). Therefore, the difference in dune shape is not strictly reflecting the difference between river and tidal environments, but rather the complex interaction between dune morphology, sediment

properties and hydrodynamics. It is therefore likely that a range of shapes are found depending on these specific morphodynamic conditions.

The influence of lee side morphology and how it fits within the coupling of feedbacks in the morphodynamic triad has not yet been systematically studied. The aim of this work is therefore to characterise flow properties (velocities and turbulence) in unidirectional flow over low and high-angle dunes with their steepest slope close to the crest or close to the trough using

numerical experiments. We hypothesise that the presence, size and length of the flow separation and turbulent wake vary depending on the value and position of the maximum angle along the lee side.

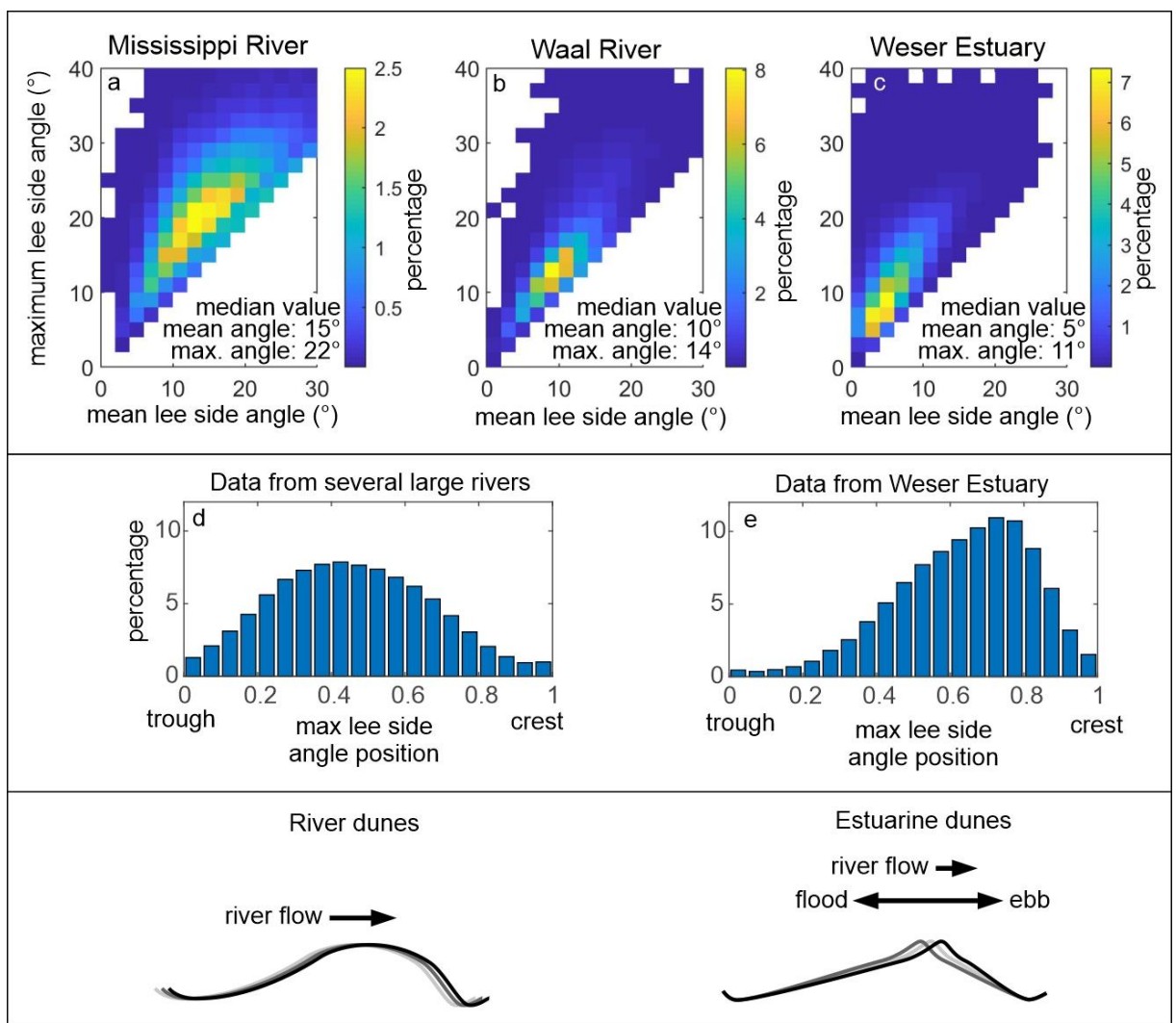

**Figure 1. Upper panel: mean and maximum lee side angles from dunes found in the Mississippi and Waal Rivers (data from Cisneros et al., 2020) and the Weser Estuary (data from Lefebvre et al., 2021). Middle panel: histogram of the position of the maximum lee side angle in six large rivers (data from Cisneros et al., 2020) and in the Weser Estuary (data from Lefebvre et al., 2021). Lower panel: schematic representation of the shape of dunes in large rivers and estuaries (after Lefebvre et al., 2021)**

## 2 Methods

### 2.1 Model description

Delft3D (Deltares, 2014) is a process-based open-source integrated flow and transport modelling system. In Delft3D-FLOW the 3D non-linear shallow water equations, derived from the three-dimensional Navier-Stokes equations for incompressible free surface flow, are solved. In order to capture non-hydrostatic flow phenomena such as flow separation and recirculation on the lee of dunes, the non-hydrostatic pressure can be computed by using a pressure correction technique: for every time step, a hydrostatic step is first performed to obtain an estimate of the velocities and water levels; a second step, taking into account the effect of the non-hydrostatic pressure, is carried out to correct the velocities and water levels such that continuity is fulfilled (Deltares, 2014).

The Delft3D modelling system has been used to set up a two-dimensional vertical (2DV) numerical model using the non-hydrostatic pressure correction technique to simulate horizontal and vertical velocities, turbulent kinetic energy (TKE) and bed shear stresses above fixed bedforms. The model has been calibrated and validated against laboratory flume experiments over idealised high-angle dunes (Lefebvre et al., 2014a) and verified against field data over natural tidal dunes (Lefebvre et al., 2014b). The same numerical model was used here to simulate flow over dunes with varying morphologies. The simulations were performed on a 2DV plane Cartesian model grid over a fixed bed (i.e. no sediment transport) composed of 10 similar bedforms. The following conditions were prescribed constant in time at the lateral open boundaries of the model domain: a logarithmic velocity profile at the upstream boundary, and a water surface elevation of 0 m at the downstream boundary. The bed roughness was set as a uniform roughness length $z_0 = 0.0001$ m. The dune height and length, the water depth and the vertical and horizontal grid size were kept similar for all simulations. The horizontal grid size was set as dx = 0.09 m (271 grid points per dune). A non-uniform vertical grid size, stretched in the vertical direction with fine spacing near the bed (dz1) and coarser spacing in the water column (dz2), was used; dz1 = 0.044 m between the trough position and the height of the crest + 5 dz1, which gradually increased to dz2 = 0.48 m within the remaining water column, resulting in 42 layers being used in each simulation. The time step dt was set to 8.33 $10^{-6}$ s (0.0005 min) following a Courant Friedrich Lewy criterion CFL = dt $\sqrt{}$ (g h) / dx < 10, where dt is the time step (seconds), g is the acceleration due to gravity (m s$^{-2}$) and h is the water depth (m). Since the z-model was used, the following condition also applies: dt ≤ dx / |u| where |u| is a characteristic value of horizontal velocities (m s$^{-1}$) (Deltares, 2014). A uniform background horizontal viscosity of $10^{-3}$ m$^2$ s$^{-1}$ and background vertical eddy viscosity of 0 m$^2$ s$^{-1}$ were set. A k-ε turbulence closure model (Uittenbogaard et al., 1992) was used.

### 2.2. Model experiments

A total of 88 simulations (Table 1) were carried out to test the influence of lee side morphology on unidirectional flow velocities, separation zone, turbulent kinetic energy and bed shear stress. Specifically, the influence of the maximum lee side slope position (closer to the dune crest or trough) was tested but not the shape of the stoss side or the overall shape of the crest.

For all simulations, bedform height ($H_b$ = 0.89 m) and length ($L_b$ = 24.4 m), water depth (h = 8 m) and mean flow velocity (0.8 m s$^{-1}$) were kept similar. The values were chosen based on typical dune dimensions in large rivers (Cisneros et al., 2020). The stoss side followed a cosine shape and the lee side was made of either a line (straight lee side) or three lines (complex lee side) (Figure 2). The straight lee side experiments were made with lee side angles varying from 5° to 30°, in increments of 5°. For each mean lee side angle ($\alpha_{mean}$), simulations were done with the lee side composed of three segments: a steep portion where the maximum angle ($\alpha_{max}$) was fixed (Table 1) and upper and lower lee sides which had angles adjusted so that the mean angle would be between 5 and 30°, in increments of 5°. The steep portion height was one third of the bedform height ($H_{sf}$ = $H_b$ / 3 = 0.3 m). For each maximum lee side angle, 4 configurations were tested, with the position of the steep portion varying from close to the crest to close to the trough (Figure 2).

A distinction was made between sharp profiles and smooth profiles. For some experiments (sharp profiles), the bed profiles were left as created with a cosine stoss side and straight or complex lee sides, with sharp variations between each section. However, most experiments were made with smooth profiles in order to mimic natural slopes: the profiles were first created from straight lines, and then smoothed using a 5-point smoothing averaged window. The smoothing produced mean angles that were lower than what was set up (Table 1). However, the maximum angle was not smoothed and stayed at the given value (Figure 2).

## 2.3. Model output analysis

From the simulation results, the horizontal and vertical velocities and the TKE are investigated above the 7[th] bedform (from a total of 10 bedforms) in order to characterize equilibrium conditions that are not perturbed by entrance and exit conditions. The position and size of the flow separation zone, when present, is calculated following the method detailed in Lefebvre et al. (2014a): the flow separation line delimitates the region in which the flow going upstream (i.e. negative horizontal velocity) is compensated by flow going downstream. The length of the flow separation zone ($L_{FSZ}$) is the horizontal distance between the separation point and the reattachment point. The relative flow separation length ($rL_{FSZ}$) is the flow separation length divided by the dune height ($H_b$). Because Delft3D uses the Reynolds-averaged Navier–Stokes equations, it is not possible to model intermittent flow separation zone; only permanent flow separation can be simulated and is considered here. The shear layer is visualised as a high vertical gradient of streamwise velocity (du/dz) reflecting a rapid change in velocity between two flow regions (Kwoll et al., 2016; Venditti, 2007).

The mean and maximum TKE ($TKE_{mean}$ and $TKE_{max}$) over the 7[th] bedform are computed as indicators of the overall turbulence produced and dissipated over each dune shape. In the literature, the wake over bedforms has been defined in different ways: following definitions of wake behind cylinders, as a momentum deficit (i.e. reduced velocity) downstream of the bedform crest (Wiberg and Nelson, 1992; Nelson et al., 1993; Maddux et al., 2003) or by its position, i.e. a specific height or water depth (e.g. 0.1 to 0.5 depth), often in cases when velocity profiles are not available at high resolution above the bedforms (Mclean

et al., 1994; Bennett and Best, 1995; Mclean et al., 1999). More recently, the wake has been defined as a region of high turbulence, using turbulence intensity and TKE above a constant threshold value (Venditti, 2007), TKE more than 70% of the

maximum TKE (Lefebvre et al., 2014a; Lefebvre et al., 2014b), high Reynolds shear stress (Kwoll et al., 2016), or percentage of quadrants 2 and 4 events (Unsworth et al., 2018). In the present work, we define the wake also based on turbulence and therefore will refer to it as "turbulent wake". The turbulent wake is the region where TKE is more than twice the average TKE above a flat bed with similar hydrodynamic conditions. For this, a simulation was carried out with a flat bed (depth 8 m) but all other parameters (e.g. domain length, boundary conditions, roughness length etc.) kept similar to the simulations with

dunes. The average TKE for this flat bed simulation is 0.0019 $m^2$ $s^{-2}$ and therefore, a threshold value of 0.0038 $m^2$ $s^{-2}$ is used to outline the contour of the wake in the dune simulations. As in previous work, this definition lacks a physical basis as the threshold value is defined quite arbitrarily. However, the turbulent wake contour above straight angle-of-repose dunes corresponds well to what is expected from the literature and it allows us to highlight changes in TKE intensity and spatial distribution between the different configurations. The length of the wake is calculated as the horizontal length between the

beginning and the end of the wake.

The bed shear stress is calculated by Delft3D using the law of the wall (Deltares, 2014). In the z-layer, strong variations of the vertical cell size affect the calculation of the bed shear stress. In order to improve accuracy and smoothness of the computed bed shear stress, the local remapping of near-bed layer is used (Platzek et al., 2014). However, due to the high resolution of the grid used here, some distortions are still seen. Therefore, we use a 10-point smoothing average to produce a smooth bed

shear stress profile.

**Table 1. Summary of bedform dimensions used for the numerical experiments. For all simulations: water depth h = 8 m; mean velocity u = 0.8 m s$^{-1}$; bedform height $H_b$ = 0.89 m and bedform length $L_b$ = 24.4 m. Note that the mean lee side angle values were fixed at 5, 10, 15, 20, 25 or 30°. However, due to horizontal resolution for the sharp profile (see also Appendix A1) and the smoothing for the smooth profiles, the mean lee side angles are lower than initially set. The maximum angle stayed as fixed (Figure2). See Figure**
**2 for a representation of the configurations**

| | | Straight lee side | | | | | |
|---|---|---|---|---|---|---|---|
| **Sharp profiles** | Mean lee side angle (°) | 5.0 | 9.9 | 14.6 | 19.5 | 24.2 | 28.7 |
| | | Complex lee side | | | | | |
| | Mean lee side angle (°) | 4.9 | 10.0 | | 20.2 | | |
| | Maximum lee side angle (°) | 10, 25 | 20 | | 30 | | |
| | Configurations for each maximum angle | 1-4 | 1-4 | | 1-4 | | |
| | | Straight lee side | | | | | |
| **Smooth profiles** | Mean lee side angle (°) | 4.9 | 9.3 | 13.5 | 17.5 | 21.2 | 24.7 |
| | | Complex lee side | | | | | |
| | Mean lee side angle (°) | 4.9 | 9.4 | 13.7 | 17.5 | 21.2 | |
| | Maximum lee side angle (°) | 10, 15, 20, 25, 30 | 15, 20, 25, 30 | 20, 25, 30 | 25, 30 | 30 | |
| | Configurations for each maximum angle | 1-4 | 1-4 | 1-4 | 1-4 | 1-4 | |

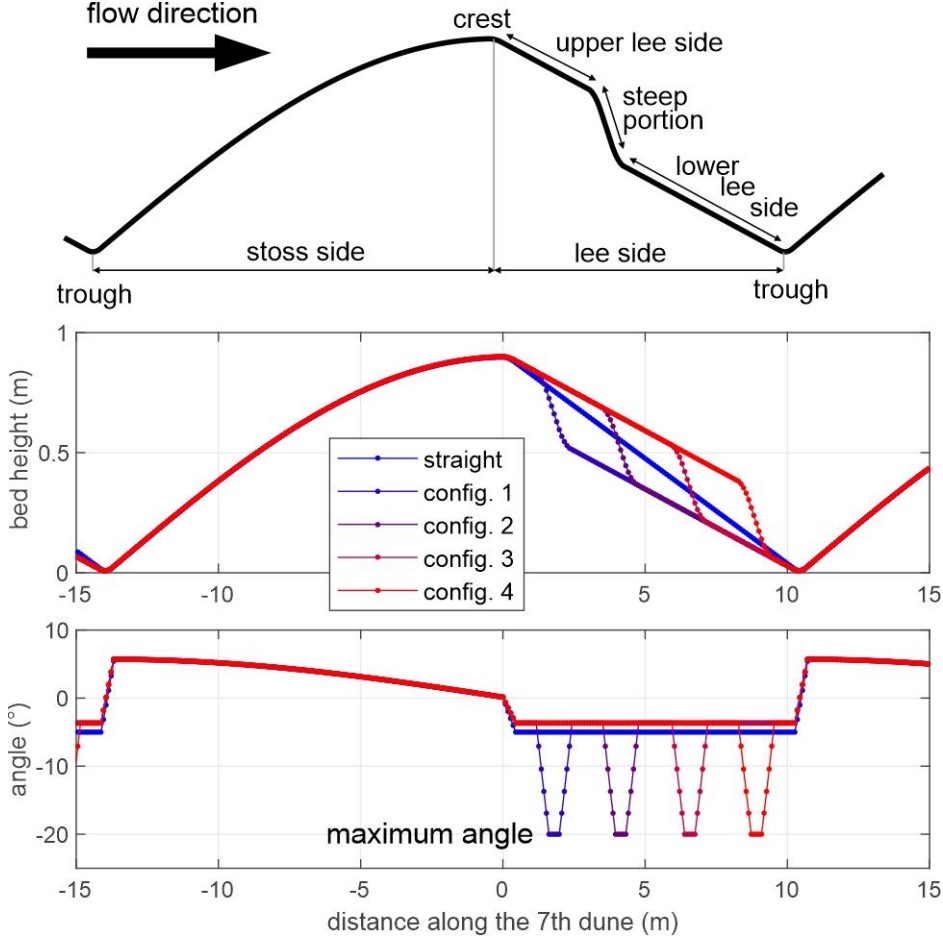

**Figure 2. Example of the different configuration tested, here for a mean lee side angle of 4.9°, a maximum angle of 20° and a smooth profile**

## 3 Results

The results from all the simulations (Figure 3) show that the relative flow separation length ($rL_{FSZ}$) and the mean TKE ($TKE_{mean}$) generally increase with increasing mean lee side angle ($\alpha_{mean}$). Both are linearly related to mean lee side angle (Figure 3a and 3d): $rL_{FSZ} = 0.17\ \alpha_{mean} - 0.67$ ($R^2 = 0.70$) and $TKE_{mean} = 0.00004\ \alpha_{mean} - 0.0009$ ($R^2 = 0.87$). They also generally increase as a function of the maximum lee side angle, but with a wide spread in the data (Figure 3b and 3e). There does not seem to be any clear pattern which relates the flow separation length or the mean TKE to the relative position of the maximum lee side angle (Figure 3c and 3f). The relative length of the turbulent wake shows strong variations and does not seem to be linearly related to mean or maximum lee side angle or its position. A distinct overall trend of relative flow separation length,

TKE and relative turbulent wake length as a function of a combination of mean lee side angle and maximum lee side angle
value and position could not be identified. There is therefore is no influence of the combined mean angle and maximum angle
value and location on flow properties. The maximum TKE above the 7th dune is linearly related to the mean TKE above the
7th dune ($R^2 = 0.95$, Appendix A2). Therefore, the trends described for $TKE_{mean}$ are also seen for the $TKE_{max}$. Although some
differences are seen between the smooth and sharp profiles, there is no systematic difference (Appendix A3).

Based on the presence and size of a flow separation zone, the shear layer and the relative length of the wake, and how they
vary depending on mean and maximum angles, it is useful to make a distinction between mean lee side less than 10° (low-
angle dunes), between ca. 10° to 17° (intermediate-angle dunes), and more than ca. 17° (high-angle dunes). Therefore, we
describe flow properties and how they are affected by the value and position of the maximum angle based on these three
categories in the next sections.

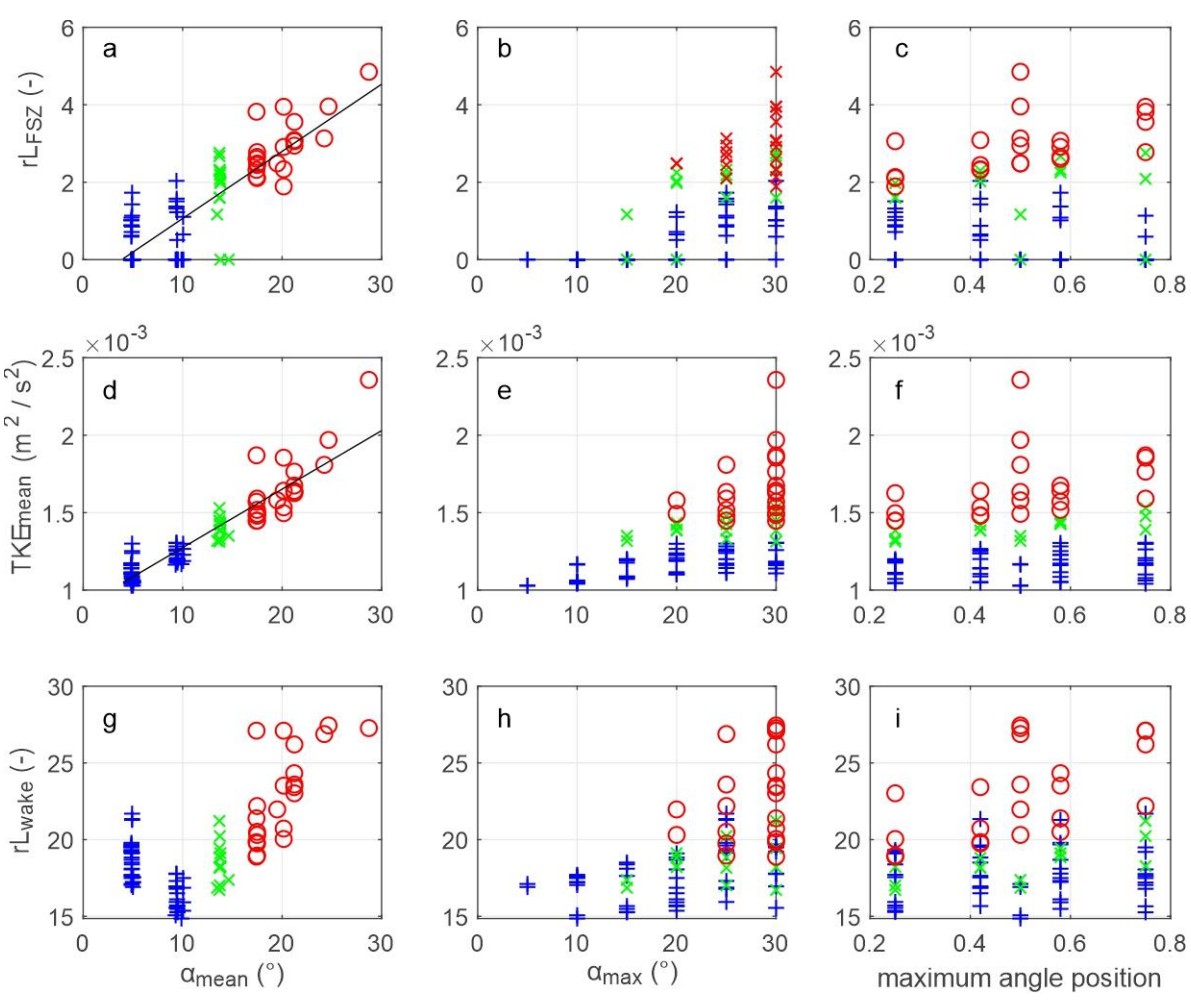


**Figure 3. Relative flow separation length (rL$_{FSZ}$), mean Turbulent Kinetic Energy (TKE$_{mean}$) and relative length of the turbulent wake (rL$_{wake}$) as a function of mean lee side angle ($\alpha_{mean}$) (a, d, g), maximum lee side angle ($\alpha_{max}$) (b, e, h) and the position of the maximum lee side angle (a position of 0.5 indicates a straight lee side) (c, f, i). The blue plusses show results from dunes with mean lee side < 10°, green crosses for mean lee sides between 10 and 17° and red circles for mean lee side > 17°**


### 3.1 Low-angle dunes

Low-angle dunes (mean lee side angle < 10°) represent 89% of the dunes measured in the Weser by Lefebvre et al. (2021) and 41% of dunes measured in six large rivers by Cisneros et al. (2020). Flow and turbulence patterns over low-angle dunes are illustrated by Figure 4, which shows dunes with a mean lee side of around 5° and a maximum angle of 10°, the most common

configuration in the Weser Estuary. As typically observed over dunes, the horizontal velocity is highest above the crest and lowest above the trough. Vertical velocity shows downward flow above the lee side, with the strongest downward flow observed above the steep portion, and upward flow above the stoss side. There is generally no flow separation over dunes with a mean lee side angle of 10° or less. However, if the maximum angle is at least 20° and situated close to the trough, a small flow separation may develop over the steep portion (Figure 5). For example, dunes with a mean lee side of ca. 10° and a

maximum lee side of 20° will not have a flow separation for config1 and config2 (where the maximum angle is positioned in the upper part of the lee side) but will have a small flow separation for config3 and config4 (where the maximum angle is positioned in the lower part of the lee side). It should be noted that dunes with a lee side angle of less than 10° but a maximum lee side angle of 20° or more represents only 2.75% of all dunes measured by Cisneros et al. (2020) in large rivers and 1.70% of dunes measured in the Weser Estuary by Lefebvre et al. (2021). Therefore, they are not commonly observed.

Over low-angle dunes, the shear layer stays close to the bed and is the thickest above the upper lee side and the steep face. The maximum TKE is situated close to the trough independently of the steep face position. The turbulent wake generally starts over the steep face and extends downstream down to a distance of ca. 5 H$_b$ after the trough. Therefore, although the mean and maximum TKE are strongest for steep faces closest to the trough, the turbulent wake is longest for steep faces close to the crest (Figure 4).


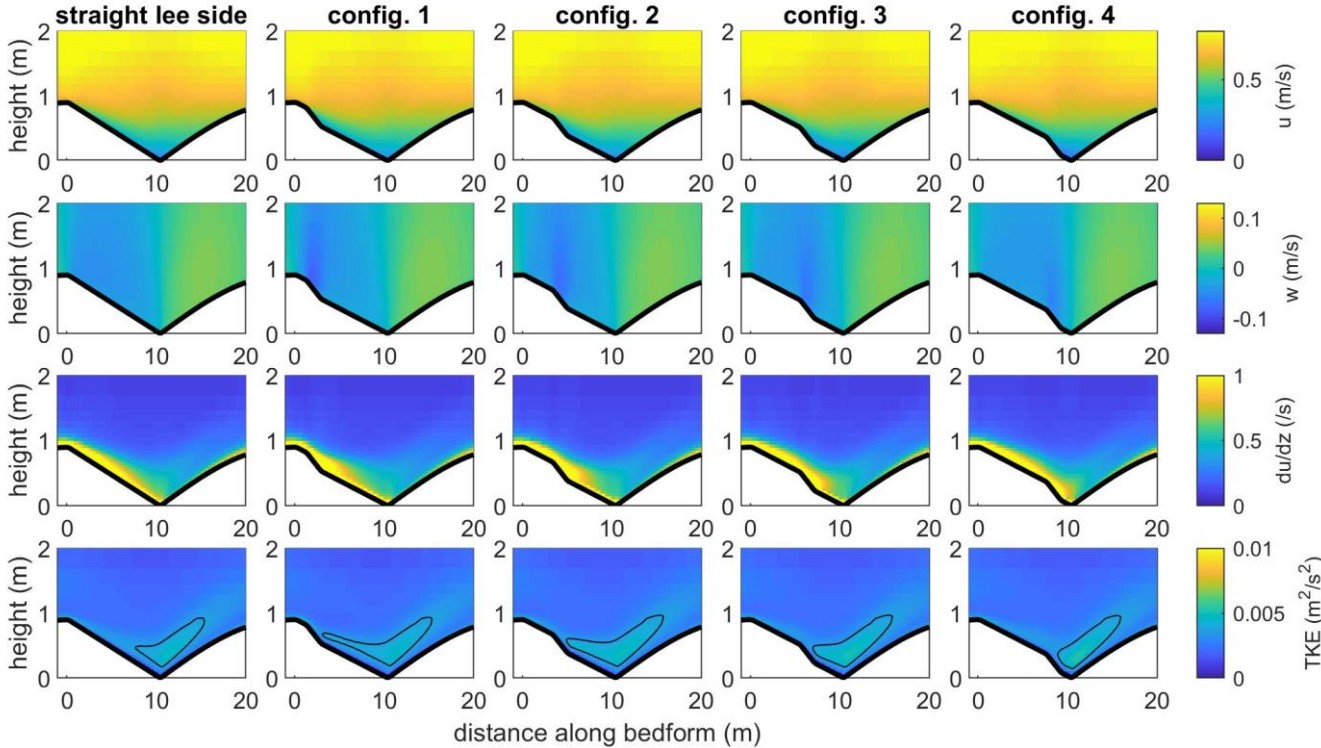

**Figure 4. Horizontal and vertical velocities (u and w), vertical gradient of streamwise velocity (du/dz) from which the shear layer can be seen and TKE above bedforms with a mean lee side of 4.9° and maximum angle of 10° for the straight lee slope and the four lee shape configurations tested. The black lines on the bottom figures show the contour of TKE>0.0038 m²/s² to highlight the position of the turbulent wake**


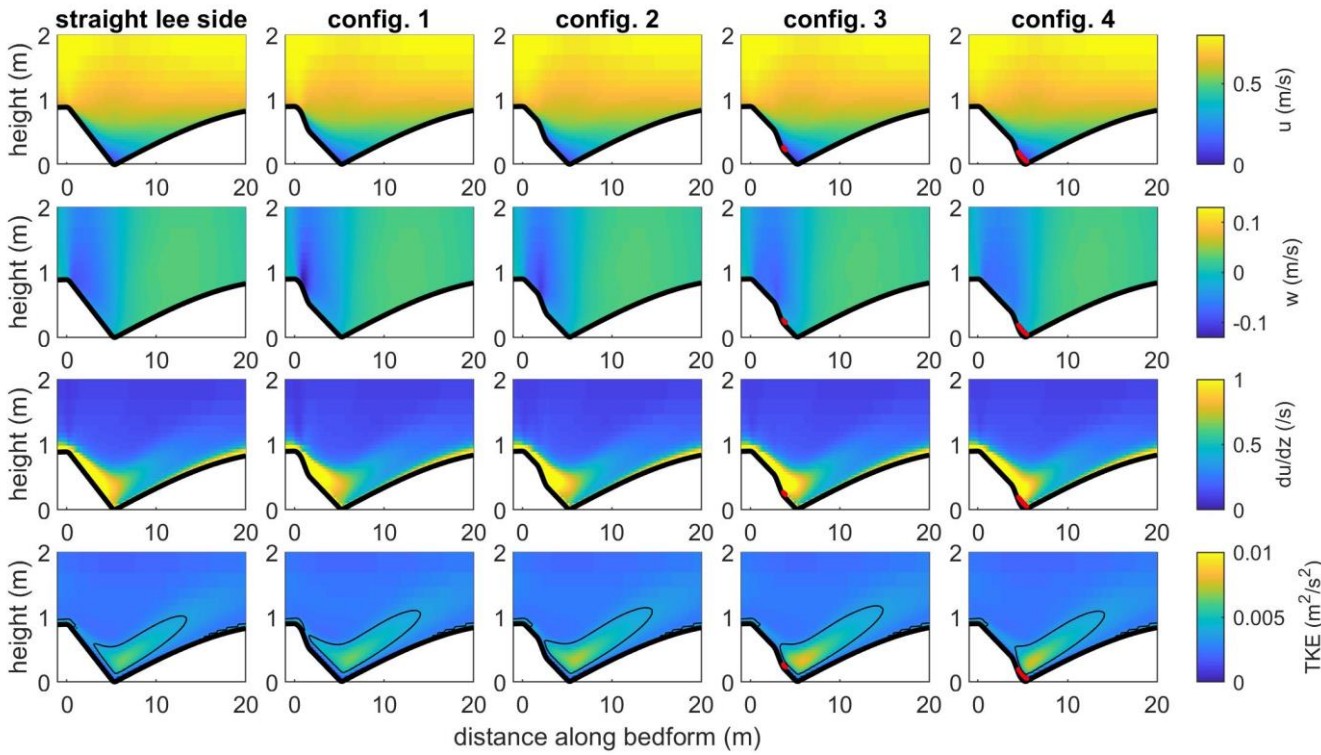

**Figure 5. Horizontal and vertical velocities (u and w), vertical gradient of streamwise velocity (du/dz) from which the shear layer can be seen and TKE above bedforms with a mean lee side of 9.4° and maximum angle of 20° for the straight lee slope and the four lee shape configurations tested. The thick red lines show the upper limit of the flow separation zone. The black lines on the bottom figures show the contour of TKE>0.0038 m²/s² to highlight the position of the turbulent wake**

### 3.2 Intermediate-angle dunes

Intermediate-angle dunes (defined here as mean lee side angles between 10° and 17°) represents 11% of the dunes measured in the Weser by Lefebvre et al. (2021) and 43% of dunes measured in six large rivers by Cisneros et al. (2020). From results of simulations over intermediate-angle dunes, it is difficult to find distinctive trends in terms of the presence and size of flow separation and TKE patterns with relation to the maximum angle value and its position. This is likely due to the fact that the model can only simulate permanent flow separation whereas intermittent flow separation is most likely to occur over intermediate-angle dunes. In general, a small permanent flow separation is simulated, although not systematically. For example, a flow separation is often absent over dunes where the maximum angle is close to the crest (e.g. config1 in Figure 6 but also config2 in some other cases) and / or for sharp profiles. Interestingly, flow separation does not always start at the brink point but is centered over the trough. As a result, the size of the flow separation, when present, varies only little with the position of the maximum slope. Moreover, over intermediate-angle dunes, the shear layer extends from the dune crest and down the lee side reaching just above the dune trough regardless of maximum angle location. It becomes thicker moving down the lee side until detaching near the location of flow separation. The maximum TKE is found close to the trough. The turbulent

wake is situated over the trough, independently of the maximum angle position. It is longer than over low-angle dunes and extends into water column above the dune height, but its size does not vary significantly depending on the position of the maximum angle for a given mean angle.

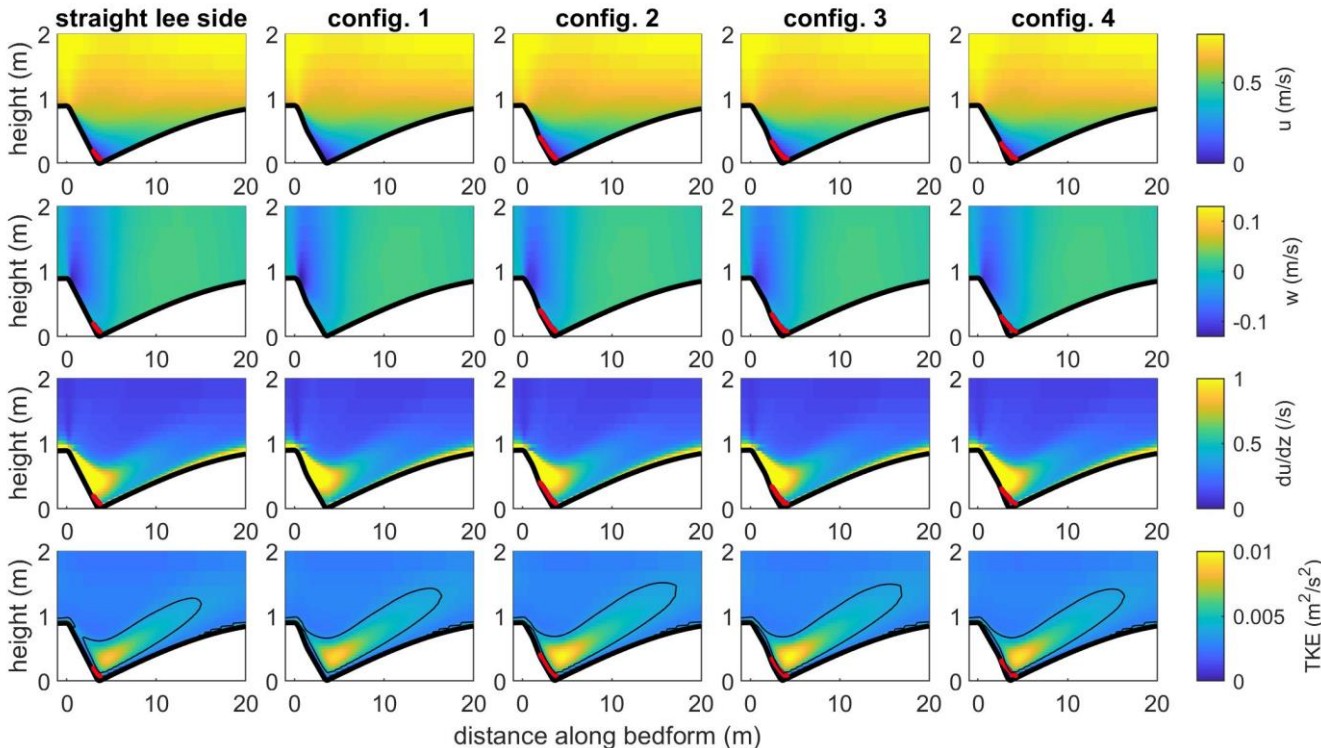

**Figure 6. Horizontal and vertical velocities (u and w), vertical gradient of streamwise velocity (du/dz) from which the shear layer can be seen and TKE above bedforms with a mean lee side of 13.7° and maximum angle of 20° for the straight lee slope and the four lee shape configurations tested. The thick red lines show the upper limit of the flow separation zone. The black lines on the bottom figures show the contour of TKE>0.0038 m²/s² to highlight the position of the turbulent wake**

**3.3 High-angle dunes**

High-angle dunes (defined here as dunes with mean lee side angles over 17°) represents only 0.03% of the dunes measured in the Weser by Lefebvre et al. (2021) and 16% of dunes measured in six large rivers by Cisneros et al. (2020). According to our simulations, a permanent flow separation is always observed above high-angle dunes. The flow generally separates over the steep portion and reattaches shortly after the trough (Figure 7). Therefore, flow separation is longer for maximum angles 250 situated close to the crest than for those situated close to the trough. The highest downward velocity is always situated just after the crest, independently of the maximum angle position. The shear layer detaches from the lee side along the upper flow

separation line. Significant differences are observed for the turbulent wake depending on the maximum angle position: it is especially strong (i.e. high TKE intensity) and spatially developed for steep portions close to the crest, so much so that a stacked wake (i.e. a turbulent wake which extends over the crest onto the next dune, as commonly observed over angle-of-repose dunes (Unsworth et al., 2018)) is seen (Figure 7). As the position of the steep portion is getting closer to the trough, the turbulent wake decreases in size and intensity.

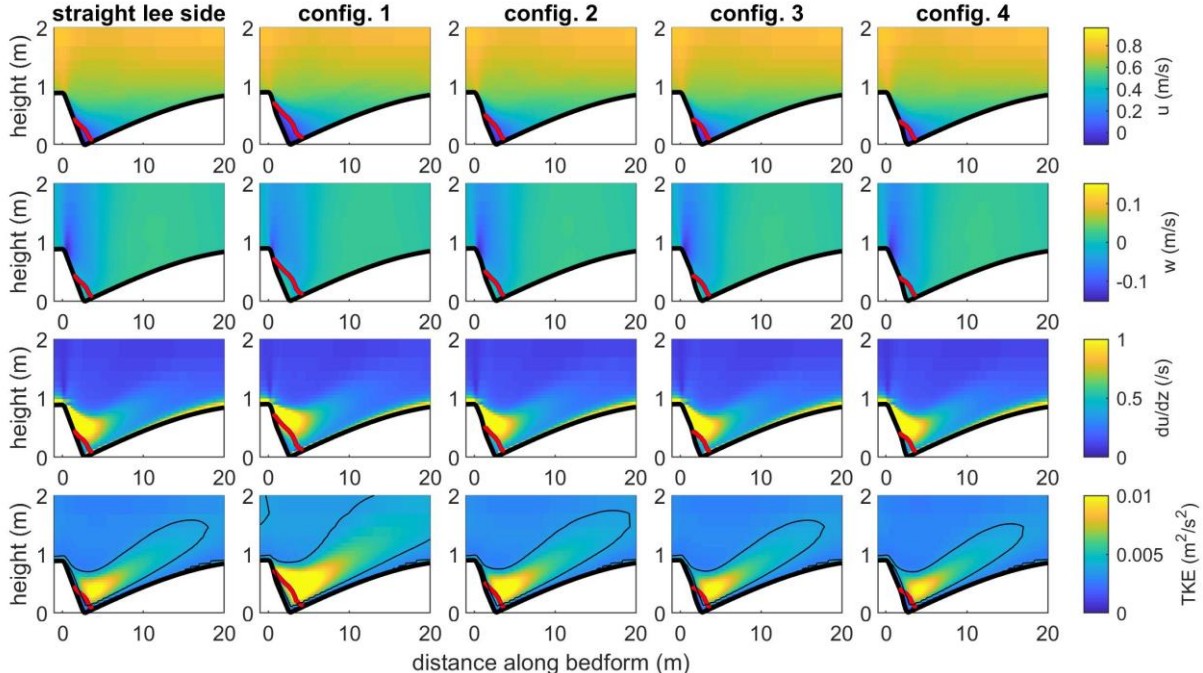

**Figure 7. Horizontal and vertical velocities (u and w), vertical gradient of streamwise velocity (du/dz) from which the shear layer can be seen and TKE above bedforms with a mean lee side of 17.5° and maximum angle of 30° for the straight lee slope and the four lee shape configurations tested. The thick red lines show the upper limit of the flow separation zone. The black lines on the bottom figures show the contour of TKE>0.0038 m²/s² to highlight the position of the turbulent wake**

### 3.4 Bed shear stress

The bed shear stress variations above the dunes are influenced by the dune shape (Figure 8). Bed shear stress generally increases along the stoss side and reaches a maximum at the crest. It then decreases along the lee side and reaches a minimum above the trough. However, this decrease is not linear even when the lee side is straight. Instead, there is a strong decrease over the first half of the lee side, and a slower decrease or stabilisation in the lower half. It should be noted that the bed shear stress over a 30° lee side angle reaches slightly negative values (minimum value of -0.01 N m$^{-2}$ over a straight lee side) due to flow separation, but stays relatively high for a 10° lee side angle (minimum value of 0.21 N m$^{-2}$ over a straight lee side). The

maximum lee side angle position has an effect which is observed mainly over low-angle lee sides. If the maximum angle is close to the crest, the bed shear stress decreases strongly over the steep portion, increases gently over the lower lee side, with a little dip over the trough and a strong increase over the stoss side. If the maximum angle is situated towards the trough, bed shear stress decreases slowly over the upper lee side before a sudden decrease over the steep portion reaching a minimum over the trough. The value of the minimum bed shear stress also varies depending on the maximum angle position. For a mean angle of around 5° and a maximum angle of 30°, minimum bed shear stress is 0.03 N m$^{-2}$ if the maximum angle is close to the crest and -0.01 N m$^{-2}$ if the maximum angle is close to the trough (presence of a small flow separation). Therefore, the position and value of the minimum bed shear stress is impacted by the position of the maximum angle.

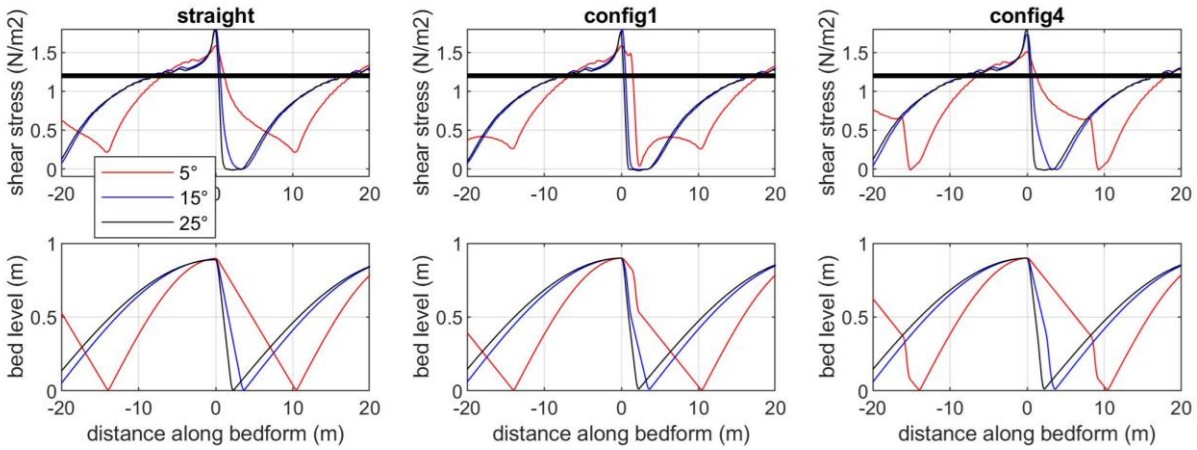

**Figure 8. Bed shear stress and bed level for bedforms with a mean lee side of 5°, 15° and 25° and with a straight lee side (left panel), or a maximum angle of 30° close to the crest (config1, middle panel) or close to the trough (config4, right panel). The thick black line on the upper plots shows the critical shear stress for bed load transport**

## 4 Discussion

### 4.1 Low, intermediate and high-angle dunes

Our results show that the mean lee side angle has the strongest control over flow separation and turbulence over dunes, with a secondary influence of the position and value of the maximum lee side angle depending on mean lee side angle values. Following these results, we propose a distinction among three types of dunes: low-angle dunes, intermediate-angle dunes, and high-angle dunes. The results of the 88 simulations carried out indicate that the boundaries between these dune types are at 10° and 17°. Additional simulations were done in order to verify these angles (Appendix A4). The simulation results confirmed that dunes with a mean lee side of 12°, 15° and 16° belong to intermediate-angle dunes, with an "irregular" pattern of flow separation and turbulence. They also showed that an increase in relative dune height (varied within the limits given by Bradley

and Venditti (2017)) did not have a strong influence on flow properties. Finally, flow over a mean lee side of 18° showed properties similar to high-angle dunes. Therefore, these additional simulations confirmed that with the settings used in the present study (fixed bedform height and length, only a small variation in bedform relative height, fixed bed roughness and flow velocity), the boundary between low and intermediate-angle dunes is at 10° and between intermediate and high-angle dunes is at 17°. The proposed classification scheme adds precision to previous research in which only high and low-angle dunes were described, sometimes without a clear definition, and did not (explicitly) recognise intermediate-angle dunes. For example, Best (2005) refers to high-angle and low-angle dunes but did not specify the slopes at which they are differentiated. Roden (1998) defined dunes with lee side angle > 20° as steep and lee side angle <10 as low-angle dune. Kostaschuk and Villard (1996) differentiate symmetric dunes with mean lee side slopes < 8° and asymmetric dunes with slopes > 19° and Kostaschuk and Venditti (2019) classify high-angle dunes with maximum lee side angles > 24° and low-angle dunes with maximum lee side angles < 24°. These three publications therefore recognised the existence of intermediate-angle dunes but only implicitly. Best and Kostaschuk (2002) define low angle lee sides with slopes <10° but mention the "transition region for the onset of flow separation (e.g., 10° – 15°)". The boundaries between dune types in the literature varies compared to those from the present study. That is likely due to the variety of dune and flow characteristics found in a natural environment compared to our simplified numerical experiments. Other numerical experiments have already showed that some characteristics (for example bed roughness, relative bedform height or aspect ratio) can influence flow separation properties (Lefebvre et al., 2014a). Therefore, it is likely that the limits between low, intermediate and high-angle dunes are not at specific fixed values but vary around these values depending on hydrodynamic and morphodynamic conditions.

Based on our results and previous research (e.g. Best and Kostaschuk, 2002; Bradley et al., 2013; Kwoll et al., 2016; Naqshband et al., 2018; Kostaschuk and Villard, 1996), properties of each dune category can be identified (Figure 9). Over low-angle dunes, there is no flow separation, except if a very steep portion (slope > 20°) is found. Low-angle dunes generate little turbulence and are likely to induce little bedform roughness. The shear layer follows the bed and is more extended when the steepest slope is close to the trough than close to the crest. Over intermediate dunes, flow separation is intermittent. Turbulence and roughness are intermediate between low and high-angle dunes. No patterns were found between the position of the maximum angle and flow properties. Over high-angle dunes, a developing to fully developed flow separation is present, a strong turbulent flow is observed and a high bedform roughness is created. If the maximum angle is close to the crest, flow separation is longer and the turbulent wake is stronger than if the maximum angle is close to the trough. The distinction between low, intermediate and high-angle dunes is important for a range of processes, for example the evaluation of bed roughness, understanding the relation between hydrodynamics, sediment transport, and dune morphology, how dunes are identified in the depositional record, and unravelling the controlling processes leading to different lee side angle slopes and shapes.

A limitation of this study is that Delft3D uses the Reynolds-averaged Navier Stokes equations and therefore can only model permanent flow separation. This is especially important in the case of the intermediate-angle dunes, which are characterised by intermittent flow separation (Best and Kostaschuk, 2002). It is likely that over intermediate-angle dunes, the turbulent wake and the shear layer size and extent will be largely dynamic, switching between the behaviour observed for the low-angle and

high-angle dunes. A particular question arises from the observation that a flow separation is often but not systematically absent when the maximum slope is close to the crest (config1 and config2) but generally present when the maximum slope is close to the trough (config3 and config4). Our results show the limitations of studying intermediate dunes with Reynolds-averaged models such as Delft3D. We suggest that these dunes should be investigated with laboratory experiments, field measurements and Reynolds-resolving models in order to precisely characterise the case of intermittent flow separation where little information is known.

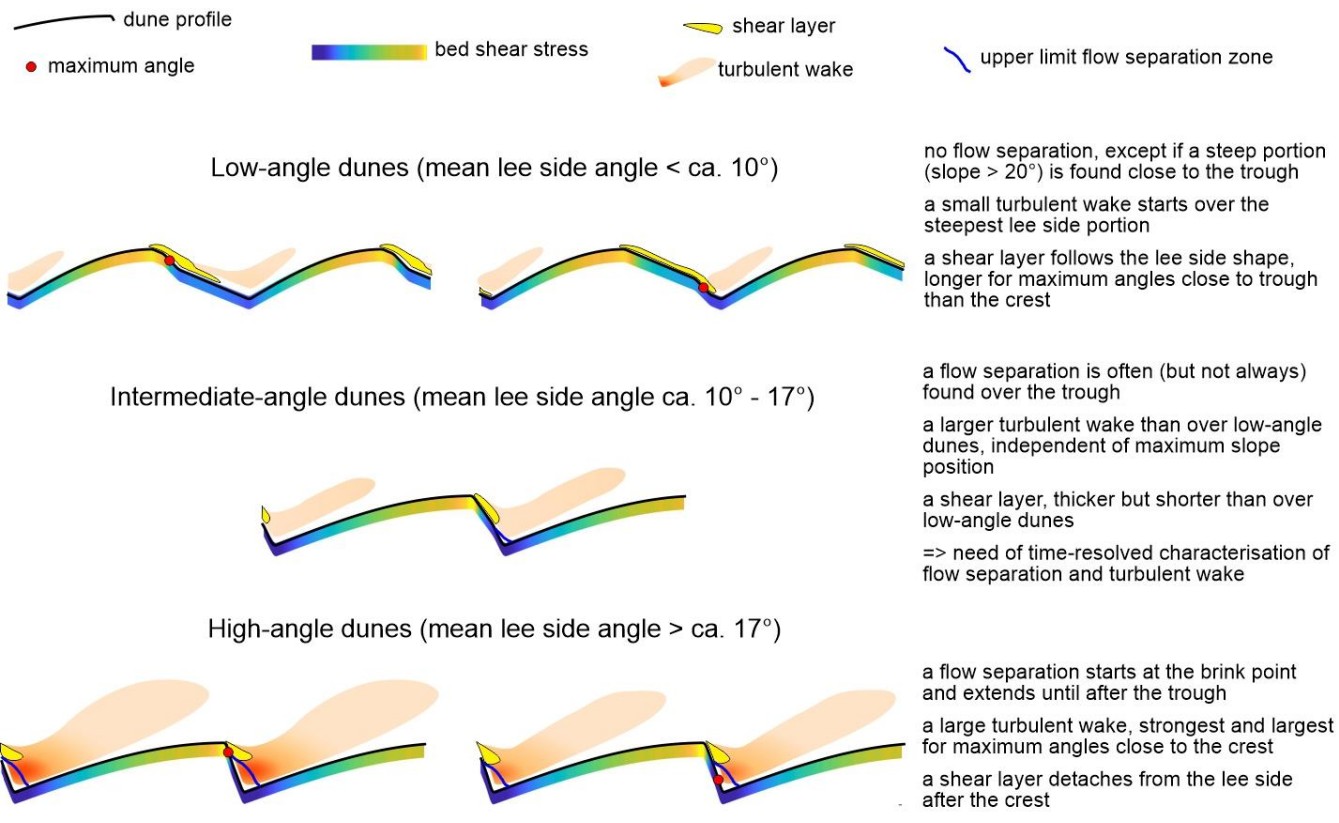

**Figure 9. Summary of the results showing the main characteristics of flow over low-angle, intermediate-angle and high-angle dunes**

## 4.2 Lee side morphology

We investigated the influence of "complex" dune morphologies, still simplified as three portions, but with a maximum angle placed towards the crest (creating a sharp crest) or placed towards the trough (creating a rounded crest). Dunes in rivers are predominantly low to intermediate-angle dunes with a rounded crest (Cisneros et al., 2020) and are similar to the config3 and config4 morphologies, which means that flow over river dunes will likely follow the results observed over those configurations.

Following this assumption, most river dunes are likely to have intermittent flow separation and a wake contained over the trough. The bed shear stress will be slowly decreasing over the upper lee side and strongly decreasing over the steep face.

Dunes in estuaries are predominantly low-angle dunes with a sharp crest (Lefebvre et al., 2021; Dalrymple and Rhodes, 1995) so they will mainly follow the results from config1 and config2. Following the modelled results over these lee side morphologies, it is likely that there is no permanent flow separation over low-angle sharp crested dunes. The turbulence will be low and diffused along a large, long turbulent wake with low TKE. The bed shear stress will have a strong decrease over the upper part of the lee side and then be stable or increase slightly over the lower lee side.

Our results show that the dune shape will affect how they interact with the flow. However, so far, the precise mean and maximum angles and the detailed shape (e.g. position of maximum angle) has been systematically quantified only for six large rivers and the Weser Estuary. Dunes on the continental shelf have been described as sharp or round crested (Van Landeghem et al., 2009; Zhang et al., 2019) but without further precision. Therefore, it is essential that the lee side shape is characterised with a precise determination of mean and maximum angles, the position of the maximum angle, if possible the size and location

of the steep face (i.e. not just the position of the maximum angle but the size of the steepest slope) and a quantification of the crest shape (between rounded and sharp) in order to precisely understand and predict the complex interaction between hydrodynamics, sediment transport and bed morphology. Since high-resolution multibeam data are now routinely collected during surveys, the description of dune shape can and should be routinely done. This would greatly help in better understand the complex interaction between bedform morphology and hydrodynamics.

In this study, we have assumed that there was only one scale of dunes. However, compound bedforms are often observed, with smaller bedforms superimposed on larger ones, usually over the stoss side. When the lee side of large dunes is low, secondary bedforms may also be found over the lee side (Zomer et al., 2021; Galeazzi et al., 2018). These secondary bedforms can have a strong influence on flow above dunes, as even if the primary dune mean lee side angle is low, the lee side of the secondary bedforms may be steep and therefore, flow separation and roughness will be created over such compound dunes. The

simulations with mean lee side 5° and maximum lee side >20° certainly show the possibility for permanent flow separation over such low-angle primary dunes if there are some steep secondary bedforms over the lee side. This has the potential to create high roughness as the wakes are mixing with one another when they are advecting downstream in the case that several steep secondary bedforms are found on the lee side of low-angle primary dunes. Further studies could therefore investigate the interaction between the turbulence and flow separation of the secondary bedforms found over low-angle dunes. However, this

does require high-spatial resolution for the secondary bedforms to be properly resolved and, if the experiments are carried out with moving sediment, the temporal scales required to observe this interaction would be related to the rate of superimposed bedform migration.

## 4.3 Potential impact on bedform roughness

Bedform roughness varies depending on lee side angle (Kwoll et al., 2016; Lefebvre and Winter, 2016). In the present study, we could not quantify directly the variations in roughness. However, bedform roughness is related to turbulence intensity over dunes (Lefebvre et al., 2016). Here, we observe that mean and maximum turbulence generally increases with increasing mean lee side angle. However, for the same mean lee side angle, there are still variations in turbulence created by the value and position of the maximum slope (Figure 3). This means that bedform roughness is likely to be affected not only by the mean

lee side angle, but also by the value and position of the maximum angle. However, no practical way has been found to characterise the roughness of low-angle dunes, and a definite relation between lee side angle of natural dune field and roughness has not yet been established. For example, no relation was found between dune lee properties and roughness variations in the river Waal in the Netherlands (De Lange et al., 2021). Therefore, it currently seems unlikely that the effect of the value and position of the maximum lee side angle can easily be considered when estimating bedform roughness of a natural

bedform field. Instead, research should focus on relating bedform statistical properties, such as the integrated slope area, roughness measurements in the field, and developing practical ways to incorporate these in model simulations. Indeed, bedform roughness variations due to bedform asymmetry has a strong impact on velocities and bedload transport calculation in tidal environments (Herrling et al., 2021)

## 4.4 Potential impact on sediment transport


The variation of bed shear stress across different dune morphologies will impact the potential for sediment transport. Here, it should be noted that the critical shear was calculated using a single sediment size and without bed slope effect (following the procedure described by Soulsby (1997)), and of course, it does not take into account the complex feedback between sediment transport and morphology variation (i.e. no sediment is being moved). Furthermore, the dune shape investigated are still simple

compared to natural dunes, especially the stoss side which is represented by a sinusoidal. The critical shear stress is therefore a simple indication for the potential to put sediment in motion along the dune but cannot be used for a full analysis of sediment transport along the dunes. This point is especially salient considering the differences in near-bed sediment transport processes between migrating and fixed dunes, specifically the presence of a dense sediment layer near the bed in migrating dunes (Naqshband et al., 2014b). For low-angle dunes (mean lee side $<10°$), the extended area of critical bed shear stresses near the

crest in case the maximum angle is near the crest (config1) implies that there is higher ability of sediment to be transported from the dune crest compared to the configuration where the maximum angle is near the trough (config4). Kostaschuk and Villard (1996) attribute crestal rounding (possibly through crestal erosion) to high near-bed velocities when bedload transport is dominating and this process would likely also occur in the config 1 case. On the other hand, there is very little difference in the bed shear stress curves for the high-angle dunes, which implies that the sediment transport potential for the different dune

morphologies will be less impacted by the location of a steep slope in these high-angle cases. The difference between the shear

stress curves from the low to the high-angle dunes requires that we account for the spatial variations in bed shear stress across the dune to better understand the sediment transport potential in these systems and the bedform evolution which can occur from these different bed shear stress patterns.

The differences in bed shear stress curves for the low-angle dunes are related to variations in velocity magnitude above the lee side, which can also be recognised in the shapes of the shear layer and turbulent wake. The magnitude of Kelvin-Helmholtz instabilities and macroturbulence above dunes have been linked to the velocity differential across the shear layer (Bennett and Best, 1995; Shugar et al., 2010) where accelerations and decelerations are attributed to locations and modes of sediment entrainment. The present simulations show differences in velocity deceleration across the crestal and lee side region depending on the maximum angle position, particularly a rapid decrease versus a gradual decrease when the steep portion is close to the crest or trough, respectively. These differences are likely to result in varying magnitudes and types of sediment transport (i.e. bedload vs. suspended load), where more rapid flow deceleration may be related to higher suspended sediment flux (Shugar et al., 2010). This shows the ways the differing flow dynamics and wake regions over complex lee side shapes may influence the along stream bed shear stress and resultant sediment transport across low-angle dunes.

The present analysis was carried out in a two-dimensional (2D) setting, whereas dunes usually have some degree of three-dimensionality. The complex interaction between bedform three-dimensional (3D) shape, flow and sediment transport has been recognised since the early work of Allen (1968) and described based mostly on idealised bedforms in physical and numerical experiments (Unsworth et al., 2020; Maddux et al., 2003; Venditti, 2007; Lefebvre, 2019; Hardy et al., 2021). For example, three-dimensionality will deflect flow over the lee side (Hardy et al., 2021) and thereby affect flow separation and turbulence properties (Lefebvre, 2019; Venditti, 2007). Furthermore, if a dune is recognised from 3D bathymetry as a 3D entity between 2 troughlines (Cassol et al., 2022; Lebrec et al., 2022), lee side angles may vary laterally along a dune. In this case, the classification may be complicated by the fact that parts of a dune may be considered for example low-angle but other parts intermediate-angle dunes.

## 5 Conclusions

Numerical simulations were carried out in order to estimate the influence of the value and position of the maximum lee side angle on flow above dunes with varied mean lee side slopes in unidirectional flow. Based on our results and previous literature, we propose a distinction between three types of dunes:

Low-angle dunes, with mean lee side slopes lower than ca. 10°. Over such dunes, there is no permanent flow separation, except if the maximum angle is more than 20° and close to the trough. The turbulent wake is generally weak, but strongest and most compact (limited spatial extent) for steep maximum angles situated close to the trough. The variations in velocity magnitude and turbulence intensity along the dune influence the bed shear stress and potential for sediment transport across different lee side shapes.

Intermediate-angle dunes, with mean lee sides of ca. 10 to 17°. Over such dunes, there is rarely a permanent flow separation but it is likely that an intermittent flow separation forms. When present, flow separation is observed over the trough,

independently of the maximum lee side angle position. These dunes should be studied systemically and in detail with laboratory experiments and eddy-resolving numerical modelling.

High-angle dunes, with mean lee sides of more than ca. 17°. Over such dunes, the flow separates at, or just downstream of, the brink point and therefore, flow separation is longest if the maximum angle is close to the crest. The turbulent wake is strong and strongest and most extensive for steep maximum slopes situated close to the crest.

The scheme introduced herein is more specific than previous schemes, which only considered low and high-angle dunes, and describes the detailed flow properties that are controlled by lee side morphology. Importantly, this new classification scheme for dune lee sides allows for a precise consideration of the interaction between dune morphology and flow. To correctly take this interaction and its consequences into account, detailed reports of dune morphology from varied environments are needed.


Appendix

## Appendix A – extra figures

**Appendix A1: comparison of flow and turbulence over flat bed and straight angle-of-repose dunes**

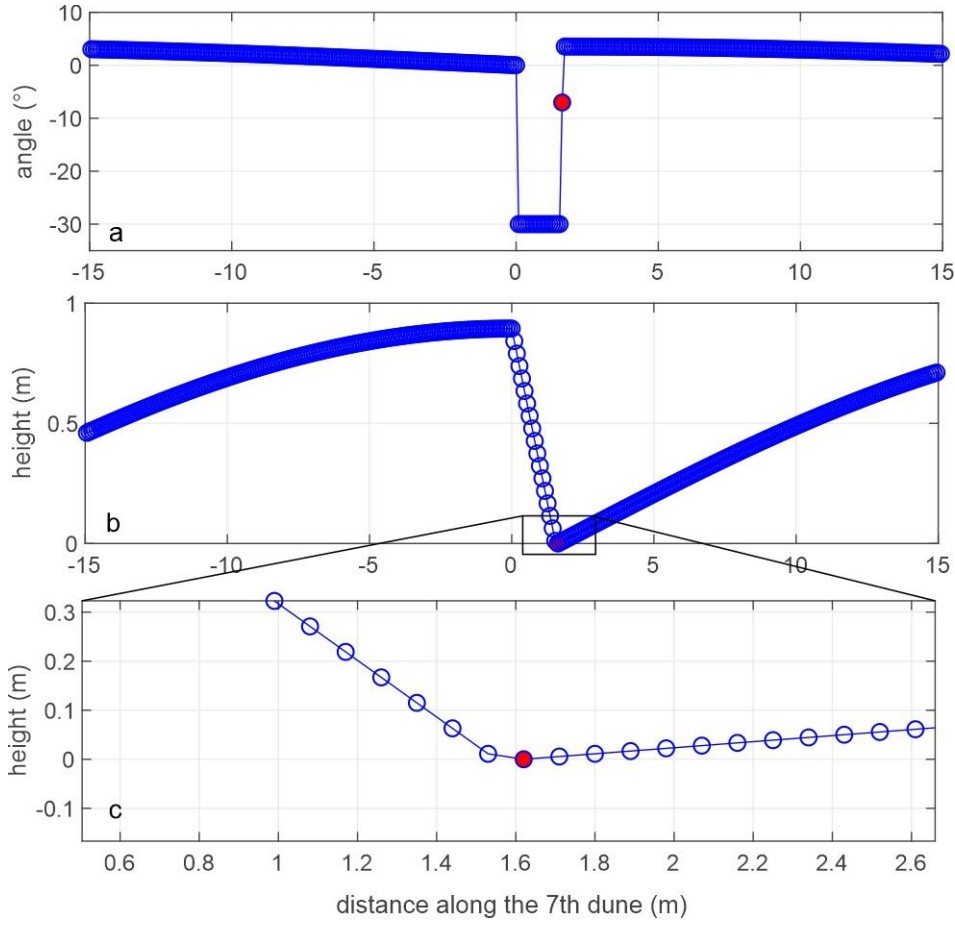

**Figure A1. Example of the angles (a) and the bed (b a close up on the trough area in c) on a straight lee side configuration with a set mean lee side angle of 30°. Because of the need to define the bedform profile on the grid, the last lee side point before the trough (marked with a red dot) is not at an angle of 30° compared to the previous point, but ca. 8°. As a result, the mean lee side angle of this dune is 28.7° instead of the prescribed 30°.**


## A2. Comparison of mean and maximum TKE over the 7<sup>th</sup> bedform for all the experiments

The maximum and mean TKE above the 7<sup>th</sup> bedform are linearly related:

$TKE_{max}$ = 12.4581 $TKE_{mean}$ - 0.0079, $R^2$ = 0.95, number of points = 88.

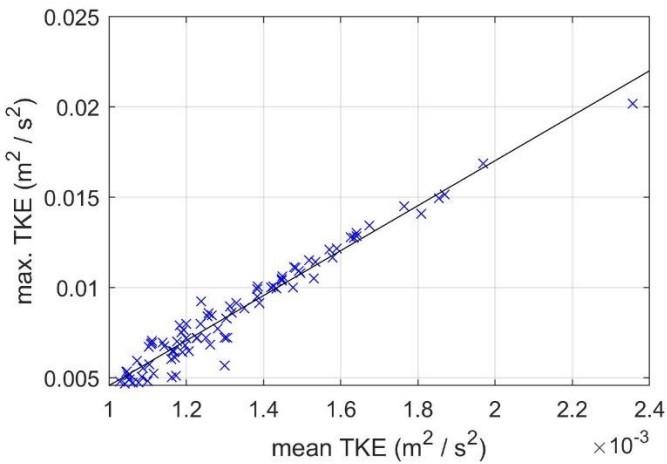

Figure A2. Mean and maximum TKE above the 7<sup>th</sup> bedform for all the simulations

 **A3. Comparison of properties over smooth and sharp lee sides**

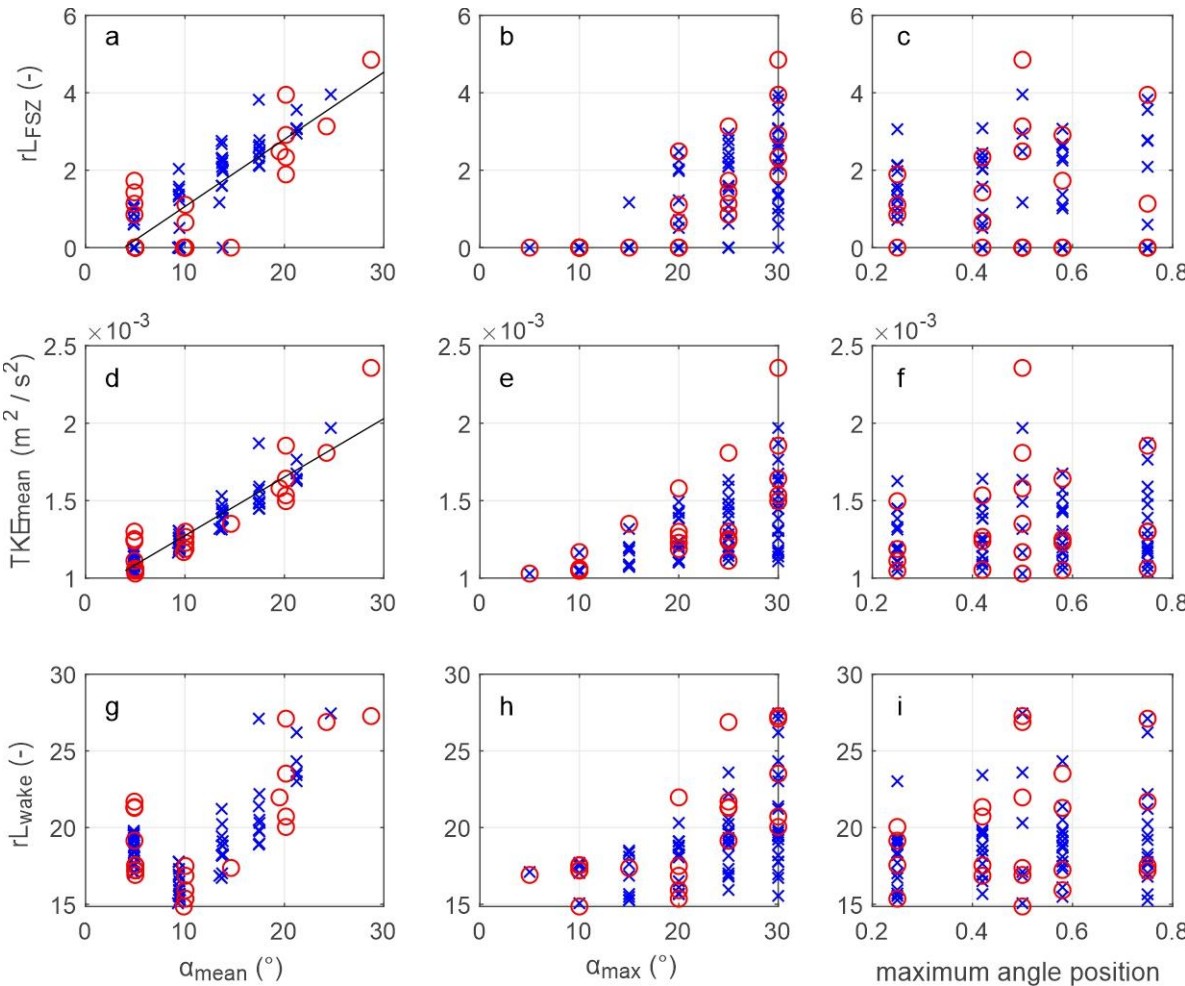

Figure A3. Relative flow separation length (rL$_{FSZ}$), mean Turbulent Kinetic Energy (TKE$_{mean}$) and relative length of the turbulent wake (rL$_{wake}$) as a function of mean lee side angle ($\alpha_{mean}$) (a, d, g), maximum lee side angle ($\alpha_{max}$) (b, e, h) and the position of the maximum lee side angle (a position of 0.5 indicates a straight lee side) (c, f, i). This figure is similar to Figure 3 from the main manuscript but the properties calculated from the sharp dunes are plotted in red circles and those from the smooth dunes in blue crosses in order to highlight the lack of systematic differences.

500

## A4. Additional simulations

Five new sets of simulations (20 simulations in total) were carried out in order to verify the limits of low, intermediate and high-angle dunes (Table 2). The results are briefly described here.

First set of simulations: dunes with a mean lee side of 13°, smoothed to 11.2°, and maximum lee side of 20°. A very small flow separation is observed over config1 and none over config2, larger one over config3 than config4. The fact that there is no direct relation between maximum angle position and flow separation size or presence, and that there is not systematically a flow separation confirms that these are intermediate-angle dunes.

Second set of simulations: dunes with a mean lee side of 17°, smoothed to 15.2°, and maximum lee side of 25°. A flow separation is found over all four configurations. However, the flow separation is in all cases concentrated over the trough and the shear layer does not detach over the crest. Furthermore, the simulation with a straight lee side of 15° does not have a flow separation. These dunes can therefore be classified as intermediate-angle dunes.

Third set of simulations: dunes with a mean lee side of 20°, smoothed to 17.5°, and maximum lee side of 25° but in 10 m water depth (instead of 8 m). The results are very similar to the similar simulation in 8 m water depth. This shows that there is not a large influence of water depth between these two values. In particular, it shows that the limit between intermediate and high-angle dunes is not shifted to higher values for smaller relative water depth.

Fourth set of simulations: a mean lee side of 18° (not smoothed) and maximum lee side of 25°. There is a flow separation is all cases, the flow separates at the brink point and reattaches shortly after the trough. The turbulent wake is strongest and largest for steep slopes close to the crest. These dunes can be classified as high-angle dunes.

Fifth set of simulations: a mean lee side of 16° (not smoothed) and maximum lee side of 23°. A flow separation is present for all four configurations. However, the size of the flow separation and the wake does not vary depending on steep slope position-. Therefore, these dunes can be classified as intermediate-angle dunes.

**Table A1. Summary of bedform dimensions used for the numerical experiments. For all simulations: mean velocity u = 0.8 m s⁻¹; bedform height $H_b$ = 0.89 m and bedform length $L_b$ = 24.4 m. For all simulations, water depth h = 8 m except for the one is italics for which the water depth was h = 10 m.**

| | **Complex lee side** | | |
|---|---|---|---|
| Mean lee side angle (°) | | 16.3 | 18.3 |
| Maximum lee side angle (°) | | 23 | 25 |
| Configurations for each maximum angle | | config1-4 | config1-4 |

Sharp profiles

| | **Complex lee side** | | |
|---|---|---|---|
| Mean lee side angle (°) | 12.0 | 15.2 | *17.5* |
| Maximum lee side angle (°) | 20 | 25 | *25* |
| Configurations for each maximum angle | config1-4 | config1-4 | *config1-4* |

Smooth profiles

Figure A4. Relative flow separation length (rL$_{FSZ}$), mean Turbulent Kinetic Energy (TKE$_{mean}$) and relative length of the turbulent wake (rL$_{wake}$) as a function of mean lee side angle ($\alpha_{mean}$) (a, d, g), maximum lee side angle ($\alpha_{max}$) (b, e, h) and the position of the maximum lee side angle (a position of 0.5 indicates a straight lee side) (c, f, i). The blue plusses show results from dunes with mean lee side lower than 10°, green crosses for mean lee sides between 10 and 17° and red circles for mean lee side > 17°. This figure is similar to Figure 3 from the main manuscript but the properties calculated from the sharp dunes

are plotted in red circles and those from the smooth dunes in blue crosses in order to highlight the lack of systematic differences.

**Code availability**

**Data availability**

The main results will be made available through https://www.pangaea.de/. Pangaea is a Data Publisher for Earth &

Environmental Science where data can be archived, published, and re-used following FAIR principles.

**Author contribution**

AL and JC conceptualised the study and planned the experiments; AL performed and analysed the numerical simulations; Experimental results were discussed by AL and JC; the original draft was prepared by AL and commented by JC; further writing, review and editing were done by AL and JC.

**Competing interests:**

The authors declare that they have no conflict of interest.

**Acknowledgements**

Alice Lefebvre is funded through the Cluster of Excellence ›The Ocean Floor – Earth's Uncharted Interface‹. Julia Cisneros is supported by the National Science Foundation under Award No. 1952844. Any opinions, findings, and conclusions or

recommendations expressed in this material are those of the author(s) and do not necessarily reflect the views of the National Science Foundation. We are very grateful to the comments from two reviewers and the editor Andreas Baas. We appreciate the open review style of Earth Surface Dynamics.

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
