# Peer review of "The influence of dune lee side shape on time-averaged velocities and turbulence"

_EGUsphere, 2023_

## Referee Comment (RC1)

[referee-annotated manuscript omitted]

---

## Author Response (AR1)

EC1: 'Editor's recommendation', Andreas Baas, 24 Apr 2023

Dear authors, the two reviews we have received are very supportive of this work and encourage the development of a revised manuscript. Reviewer #2 in particular has supplied extensive comments and suggestions that should be considered, requiring a moderate revision that will be returned to this reviewer for their evaluation. This reviewer recommends some additional simulation work that could strengthen the findings you report and so this is something you may consider exploring. Please let us know if you require more time to implement revisions (and potentially additional work) and we can easily extend the turn-around period.

Kind regards,

AB

**Dear editor, we are very happy about the two reviews. Indeed, Reviewer #2 did have a lot of suggestions, which we considered. For this, we have done some additional simulations. Thanks to these, the results and discussion could be extended and strengthened.**

**In general, we have agreed with the suggestions. There are some points which we have investigated but we have not made changes in the manuscripts. The details of our answers to the reviewers' comments are found below. Please note that there may be some small discrepancies between the text pasted in the answers and the final text as we read the whole manuscript again after accepting all the modifications in order to ensure coherence.**

**Thanks for the opportunity to have an open review process.**

**Best wishes,**

**Alice Lefebvre and Julia Cisneros**

Principal criteria

My response to the principal criteria is given in italic.

Scientific significance:

Does the manuscript represent a substantial contribution to scientific progress within the scope of Earth Surface Dynamics (substantial new concepts, ideas, methods, or data)?

*1. Excellent. There has been considerable debate in recent years about the role of dune lee side morphology in river flow - this paper uses a numerical model to provide a much-needed systematic examination of the impact of lee side configuration on velocity and turbulence over dunes.*

**Thank you!**

Scientific quality:

Are the scientific approach and applied methods valid? Are the results discussed in an appropriate and balanced way (consideration of related work, including appropriate references)?

*1. Excellent. The authors provide clear explanations of their approach and methods and stick to interpretations based on their results.*

**Thank you!**

Presentation quality:

Are the scientific results and conclusions presented in a clear, concise, and well-structured way (number and quality of figures/tables, appropriate use of English language)?

*1. Excellent to 2. Good. I provide some suggestions for clarification below.*

**Thank you for the suggestions, really appreciated**

**Access review, peer review, and interactive public discussion**

My responses are in italics.

Does the paper address relevant scientific questions within the scope of ESurf?

*Yes. The paper focuses on river dunes, an important component of river morphodynamics.*

Does the paper present novel concepts, ideas, tools, or data?

*Yes. The main novel contribution is the use of a numerical model to examine the effect of various lee side configurations on flow over dunes.*

Are substantial conclusions reached?

*Yes. The research leads to substantial conclusions that do a good job of summarizing the main results of the research.*

Are the scientific methods and assumptions valid and clearly outlined?

*Yes.*

Are the results sufficient to support the interpretations and conclusions?

*Yes. The authors do a good job of using the results of their experiments to support their interpretations and conclusions while avoiding excessive speculation.*

Is the description of experiments and calculations sufficiently complete and precise to allow their reproduction by fellow scientists (traceability of results)?

*Yes. Their approach could be used to examine other aspects of dune morphodynamics such as patterns of sediment transport over dunes.*

Do the authors give proper credit to related work and clearly indicate their own new/original contribution?

*Yes.*

Does the title clearly reflect the contents of the paper?

*No. I think the title needs to be more explicit in terms of the aspects of dune morphodynamics that lee side shape affects. For example, "The influence of dune lee side shape on time-averaged velocities and turbulence" better reflects the content of the paper.*

**We agree with you and we have changed the title**

Does the abstract provide a concise and complete summary?

*Yes.*

Is the overall presentation well structured and clear?

*Yes.*

Is the language fluent and precise?

*Generally yes. I have attached a marked copy of the manuscript with some editorial suggestions.*

**We have followed your suggestions**

Are mathematical formulae, symbols, abbreviations, and units correctly defined and used?

*Yes.*

Should any parts of the paper (text, formulae, figures, tables) be clarified, reduced, combined, or eliminated?

*Figure 2: This diagram is confusing. It is difficult to distinguish between the symbols (., \* etc) for each configuration. It might be better to use thinner lines for all the profiles.*

**I was trying to put symbols because it may help colour blind or people with visual deficiency. But I agree that this makes it very hard to see. I have changed the figure. It still largely understandable for colour-blind people (https://www.color-blindness.com/coblis-color-blindness-simulator/)**

*Lines 172-175: Perhaps I have missed something here, but I don't understand why mean lee side values of 10 and 17 degrees have been chosen as boundaries for these dune types - additional explanation is required.*

**This was rephrased into "**Based on the presence and size of a flow separation zone, the shear layer and the relative length of the wake, and how they vary depending on mean and maximum angles, it is useful to make a distinction between mean lee side less than 10° (low-angle dunes), between ca. 10° to 17° (intermediate-angle dunes), and more than ca. 17° (high-angle dunes).**"**

*lines 277-278: the sentence beginning "For example, Best (2005)...." is confusing - please clarify.*

**Rephrased into "**For example, Best (2005) refers to high-angle and low-angle dunes but did not specify the slopes at which they are differentiated.**"**

*move lines 302-308 to the beginning of section 4.1 so it is clear why the earlier use of 17 degrees has been changed to 20 degrees*

**Following comments from Reviewer 2 and additional simulations, the classification is left with 17° now, and this part was largely rewritten.**

*lines 319-320: this sentence needs clarification. "Following this assumption, river dunes, in general, are likely to have no or intermittent flow separation and a relatively strong (missing text here?) contained over the trough.*

**Rephrased into "**Following this assumption, most river dunes are likely to have intermittent flow separation and a wake contained over the trough.**"**

*end of section 4.4. The paper and the proposed classification scheme focuses on 2-dimensional dune profiles - dune morphology however is usually three dimensional so I think an additional brief paragraph should be added that considers the role of 3-morphology on dune profiles.*

**A new paragraph has been added**

"The present analysis was carried out in a two-dimensional (2D) setting, whereas dunes usually have some degree of three-dimensionality. The complex interaction between bedform three-dimensional (3D) shape, flow and sediment transport has been recognised since the early work of Allen (1968) and described based mostly on idealised bedforms in physical and numerical experiments (Unsworth

et al., 2020; Maddux et al., 2003; Venditti, 2007; Lefebvre, 2019; Hardy et al., 2021). For example, three-dimensionality will deflect flow over the lee side (Hardy et al., 2021) and thereby affect flow separation and turbulence properties (Lefebvre, 2019; Venditti, 2007). Furthermore, if a dune is recognised from 3D bathymetry as a 3D entity between 2 troughlines (Cassol et al., 2022; Lebrec et al., 2022), lee side angles may vary laterally along a dune. In this case, the classification may be complicated by the fact that parts of a dune may be considered for example low-angle but other parts intermediate-angle dunes."

Are the number and quality of references appropriate?

*Yes.*

Is the amount and quality of supplementary material appropriate?

*Yes. However, I don't understand the explanation in the caption of Figure A1 – please clarify. Also, I think the blue lines are too thick – see my comment on Figure 2 above.*

**Figure and caption changed**

"Figure A1. Example of the angles (a) and the bed (b a close up on the trough area in c) on a straight lee side configuration with a set mean lee side angle of 30°. Because of the need to define the bedform profile on the grid, the last lee side point before the trough (marked with a red dot) is not at an angle of 30° compared to the previous point, but ca. 8°. As a result, the mean lee side angle of this dune is 28.7° instead of the prescribed 30°."

**Comments from the pdf – all corrected**

l.55: the

l79. set up

l94. :

l280. differentiate

l281. Classify, maximum, maximum,

l282. implicitly

l283. define, mention

l285. and

l307. angles

l327.

l334

l335. and

l362. such as

l363.

l374. requires

l394. compact

l402. and

l403. extensive

l404-405 The scheme introduced herein, schemes, considered, detailed, that are controlled by, The, Importantly, this new classification scheme for dune lee sides

In this paper the influence of lee-slope angle and shape on flow properties is investigated through numerical modelling simulations. Insightful results are presented in outstanding figures, with findings that valuably contribute to knowledge of the interaction between bedforms and hydrodynamics. In this review, I provide suggestions for the discussion and dune classification.

Please see the attached .zip, containing:

- Review report (.doc) for general suggestions and questions to improve the manuscript.

- egusphere-2023-211_Rev.pdf with specific comments (in balloons) and text corrections (text additions).

Review report:

In this paper the influence of lee-slope angle and shape on flow properties is investigated through numerical modelling simulations. Results are presented in outstanding figures, with findings that valuably contribute to knowledge of the interaction between bedforms and hydrodynamics. However, the classification of dunes (i.e. the bounding lee-slope angles defining the classes) does not follow the results in this paper. Please see suggestions for an alternative line of thought and Figure 9.

**Reviewer's comments:**

Please see *211_Rev.pdf, for comments with the text (balloons) and text corrections (Text insertions and deletions).

- Good abstract, describing the relevance and results
- Excellent introduction, with strong background information; rationale and hypothesis.
    o In the first 100 lines of the paper, incl Table 1, I can't seem to find whether simulations are for unidirectional or both uni-dir. and oscillating flows. The only clue is 'rivers' (=uni-dir). Please add to the aim (lines 59-60): "for unidirectional flow".
    **done**
    o Some suggestions for the use of existing literature.
    **Were added**
- **Main point #1:** The paper needs a more consistent use of symbols, and equations vs. explanation in words. This balance in the text is quite unlogic in this matter, e.g. lack of explanation in words for k-ε in the model description (line 95), and on the other hand, it contains words in equations where you'd prefer symbols/an equation (line 160-161: e.g.

$L_{sr} = 0.17\ \alpha_{mean} - 0.67$, $TKE_{mean} = 0.00004\ \alpha_{mean} - 0.0009$ and appendix (line 442). Furthermore, the paper would benefit from adding units, see comment with lines 92-96).
**This was done (see details in "Comments in the pdf")**
Suggestions of how to do this:

- o Add a symbol after the description in words, then use the symbol in an equation. E.g., "The mean turbulent kinetic energy, $TKE_{mean}$, is computed as …"; and then in the results: " $TKE_{mean} = 0.00004\ \alpha_{mean} - 0.0009$".
  **Done (see section 2.3 and beginning of results)**
- o NB. These symbols could then be used on axes of result figures as well (since the text explains how these are calculated, see section 2.3 and caption Fig 3)
  **Done (see especially Figure 3)**

- Excellent reasoning for initial choices of simulation runs (section 2.2) and flow parameter analyses (section 2.3). However,
  - o The runs are all for unidirectional flow. That is okay, but the introduction suggested that crest sharpness in tidal flow conditions was also relevant to investigate. See comment above, to add to aim. P.S. Having read the paper: sharp and rounded crests are created by the position of 'the steep portion' (max lee slope segment); this became more clear when reading the results and discussion, but not so much in section 2.

    **The aims now mention that the work is focussed on unidirectional flow**
    "The aim of this work is therefore to characterise flow properties (velocities and turbulence) in unidirectional flow over low and high-angle dunes with their steepest slope close to the crest and close to the trough using numerical experiments."

    **And the fact that only the position of the maximum lee side is investigated is explicitly said, for example with the sentence at the beginning of section 2:**
    "Specifically, the influence of the maximum lee side slope position (closer to the dune crest or trough) is tested and not the shape of the stoss side or the overall shape of the crest."

  - o Line 123: no intermittend flow separation can be investigated. See Main point #2.
  - o The 5-30 degrees slopes with increments of 5 degrees will appear to be insufficient to find the angles bounding the dune classes in the discussion. See Main point #2.
    **Addressed in Main point #2**

- Results
  - o Outstanding figures in the results sections and well described in the results texts (sections 3.2, 3.3 and 3.4; which are also well structured).
    **Thanks!**
- Discussion: relevant topics in headings, however:
  - o **Main point #2:** The bounding lee-slope angles for the classes of dunes here proposed (lines 274-276), do not follow from the results presented in this paper. Please consider the following line of thought:
    - ▪ Firstly, the results show for 5 and 10°-angle dunes (Figs 4 and 5): no (nearly no) flow separation, relatively long shear layer, TKE and length

wake dependent on max slope position (lee shape). The 15° dunes (Fig 6): most discriminating observation is that flow separation, shear layer, TKE and wake length are not (hardly) correlated to lee shape (max slope position, the configurations). The 20° dunes (Fig 7) show clear effects of high-angle dunes. Thus, based on these distinctions (results): the 10° dunes would (still) fall in the class low-angle dunes (≤10°), and the 20° dunes would (already) fall in the high-angle dunes (≥20°). The intermediate dunes would fall in the approximate range >10° to <20°. This would better follow the results in this paper. The distinction into the three classes thus seems based more on previous findings in the literature than on results presented in this paper. Moreover, the classification was already proposed (i.e. defined) early in the paper. And does not differ from an existing classification in the - not so useful - explicit/implicit reasoning.

**Here, it should be stressed out that Fig. 7 shows "ca. 20° mean lee side angle dune" which, due to the smoothing, actually has a mean lee side of 17.5° (Table 1). We have adapted the figure captions to make sure that this is clear.**

- However, strictly, this paper presents av. slope angles "between 5 and 30°, in increments of 5°" (5, 10, 15° with max angles of 20° and av angle of 20° and max of 30°), but what the bounding av. lee-slope angles are, cannot be concluded from the presented results (Figures 4 – 7).  The classification proposed in this paper thus needs to search for the angles that bound the 3 classes (11 – 15° and 15-19°), like how in mapping one would have to search for the boundaries. Even though the discussion states: this would not be an exact angle (lines 307-308), the proposal of this classification is a main point of the paper, and cannot be done without searching for the bounding angles.

  **We did a total of 20 new simulations in order to bound the three classes. We now conclude that with the settings that we test, the limits are really at 10 and 17°. But we also emphasise that these are likely to be ranges instead of precise boundaries, especially in natural environments where the dune lee sides will not be made of three straight lines.**
  **We have done all this in a new paragraph specifically dedicated to this topic**

- Leaving the 17° angle (early in the paper) and replacing by 20°: does that mean that if we go back to Fig. 3 that the red circles of ~17° would then be green crosses? Why would this be a more logical distinction?
  **This is indeed not more logical. The limit is now left throughout the manuscript at 17°**

- Line 216: difficult to find distinctive trends for intermediate-angle dunes. Is this a consequence of not being able to model intermittent flow separation? (lines 122-123) I.e. in the literature the specific flow for intermediate-angle dunes. Please discuss in the discussion what the exact (or expected) consequences are for not being able to model intermittent flow separation. For example, would the independency of max lee slope position for most flow parameters for the 15° dunes be a direct consequence, and if so why? Lines 306-307 say: needs more research, but that is not sufficient here, for proposing a classification.
  **Addressed in the new paragraph in section 4.1**

- o A question that arose from Figure 6 for the 15° dune results: flow separation is absent in Config1 (whereas it is present in all other configs): why?
  **Addressed in the new paragraph in section 4.1**

- o **Figure 9** will benefit in conceptual power with a more schematic visualisation (not mimicking results figures 4 – 7), preferably without needing the text to explain.
  - ▪ Do you really need the 2 configurations? Firstly, differences are not very clear from the figure, and secondly, this could be differently visualised, for example by indicating the max slope location of flow parameters (e.g. red dot near crest or trough for shape-dependent parameter; absence of dot means no dependence). Or something alike.
    **The two configurations were left for low and high-angle dunes because the position of the maximum slope has an influence. The position of the maximum slope is now indicated with a red dot as suggested. For the intermediate-angle dune, only one morphology is presented since the maximum angle position does not have a determined influence.**

- o The paragraph on continental shelves (lines 327-337) is pure speculation (more or less a repetition of the introduction). This could bworked out adequately, but requires a fuller study of the literature. Here, therefore, I would suggest to not discuss this in the discussion, but formulate the application to continental shelves as wider implication of this work. That would turn this paragraph (weakness of the paper) into a strong point.
  **Paragraph rephrased**
  "Dunes on the continental shelf may have any type of morphology from sharp to round crested (Van Landeghem et al., 2009; Zhang et al., 2019). Our results show that the dune shape will affect how they interact with the flow. However, so far, the precise mean and maximum angles and the detailed shape (e.g. position of maximum angle) has been systematically quantified only for six large rivers and the Weser Estuary. Therefore, it is essential that the lee side shape is characterised with a precise determination of mean and maximum angles, the position of the maximum angle, if possible the size and location of the steep portion (i.e. not just the position of the maximum angle but the size of the steepest slope) and a quantification of the crest shape (between rounded and sharp) in order to precisely understand and predict the complex interaction between hydrodynamics, sediment transport and bed morphology. Since high-resolution multibeam data are now routinely collected during surveys, it can and should be routinely done. This would greatly help in better understand the complex interaction between bedform morphology and hydrodynamics."

- o With section 4.3: bed roughness: in this paper, relative lengths of wakes were used. To what extent the wakes extend into the flow (water column; exceeding dune height) is not mentioned. Would this not add to the discussion on bedform roughness?
  **From what I know of bedform roughness and turbulent wake, the relation is mainly that the turbulence intensity (mean or maximum TKE) is the main parameter related to bedform roughness. This is similar to the approach of calculating the bed shear stress as tau = 0.19 TKE**
  **I calculated the wake height above the crests for all the simulations and it is very much related to the wake length**

[Figure]

Figure. Normalised wake length as a function of normalised wake height (above the crest) for all the experiments (except those in deeper water). Note that for the two experiments with the longest wake height, the length is not correctly calculated because the wake extends over the next dune.

**For now, I haven't added anything about the wake height. I am happy to do so if you feel very strongly about it.**

o  Section 4.4: Lines 378-384 is a repetition of the results. Why not relate to the critical shear stress for incipient motion, as indicated in Fig. 8 (see earlier comment in .pdf that slope effect should be taken into account for critical shear stress?).
**It was a point that we discussed when writing the paper. We are not so keen on doing this because we feel that this would require a much deeper analysis that we are doing for this paper. It shouldn't be just the slope considered, but also the sediment grain size, the bed roughness, the mean velocity, the dune shape as a whole… Our simulations are still very simplified compared to natural conditions.**
**We added this at the beginning of the section on sediment transport**
"The variation of bed shear stress across different dune morphologies will impact the potential for sediment transport. Here, it should be noted that the critical shear was calculated using a single sediment size and without bed slope effect (following the procedure described by Soulsby (1997)), and of course, it does not take into account the complex feedback between sediment transport and morphology variation (i.e. no sediment is being moved). Furthermore, the dune shape investigated are still simple compared to natural dunes, especially the stoss side which is here represented by a sinusoidal. The critical shear stress is therefore a simple indication for the potential to put sediment in motion along the dune but cannot be used for a full analysis of sediment transport along the dunes. This point is especially salient considering the differences in near-bed sediment transport processes between migrating and fixed bed dunes, specifically the presence of a dense sediment layer near the bed in migrating dunes (Naqshband et al., 2014b)."

Also, interesting in discussing the consequences for sediment transport in this section (4.4) would be:

- does the detachment of the shear layer in higher angle cases lead to more suspended sediment (locally) i.s.o. bedload?
  **That's interesting. We have added some ideas about this comment, the previous and next comments**

  "For low-angle (mean lee side <10°) dunes, the extended area of critical bed shear stresses near the crest in case the maximum angle is near the crest (config1) implies that there is higher ability of sediment to be transported from the dune crest compared to the configuration where the maximum angle is near the trough (config4). (Kostaschuk and Villard, 1996) attribute crestal rounding (possibly through crestal erosion) to high near-bed velocities when bedload transport is dominating and this process would likely also occur in the config 1 case. On the other hand, there is very little difference in the bed shear stress curves for the high-angle dunes, which implies that the sediment transport potential for the different dune morphologies will be less impacted by the location of a steep slope in these high-angle cases. The difference between the shear stress curves from the low to the high-angle dunes requires that we account for the spatial variations in bed shear stress across the dune to better understand the sediment transport potential in these systems and the bedform evolution which can occur from these different bed shear stress patterns.

  The differences in bed shear stress curves for the low-angle dunes are related to variations in velocity magnitude above the lee side, which can also be recognised in the shapes of the shear layer and turbulent wake. The magnitude of Kelvin-Helmholtz instabilities and macroturbulence above dunes have been linked to the velocity differential across the shear layer (Bennett and Best, 1995; Shugar et al., 2010) where accelerations and decelerations are attributed to locations and modes of sediment entrainment. The present simulations show differences in velocity deceleration across the crestal and lee side region depending on the maximum angle position, particularly a rapid decrease versus a gradual decrease when the steep portion is close to the crest or trough, respectively. These differences are likely to result in varying magnitudes and types of sediment transport (i.e. bedload vs. suspended load), where more rapid flow deceleration may be related to higher suspended sediment flux (Shugar et al., 2010). This shows the ways the differing flow dynamics and wake regions over complex lee side shapes may influence the along stream bed shear stress and resultant sediment transport across low-angle dunes."

- the line of critical shear stress (Fig 8) implies that sediment will only be mobilised at the crests of dunes. However, empirical studies show that large parts of stoss sides are being eroded. Is this a consequence of the choice of initial dune shape in the simulation runs?
  **I suspect so. I think it is a combination of sediment variation along the dunes and the shape of the stoss side. The few lines that are added at the beginning of this paragraph address this.**

- Conclusions fit the current line of thought in the current text. With adjusting the proposed classification, the conclusions would have to be adjusted as well.
  **Conclusions were adjusted**
- Ref list:
  - See additional literature suggested in the comments in *_Rev.pdf
  - To the Kostaschuk & Vendetti 2019 paper, a comment was written (2020, Cisneros is one of the co-authors). Would it be wise to also use the comment?
    **The comment is mostly in response to the processes by which low angle dunes form and this citation is only being used to say that a distinction was made between low and high angle dunes. So we decided not to cite the comment.**

**Reviewer's recommendation to Editor**

I recommend publishing in ESD with revisions, taking into account the comments and textual suggestions/corrections. Main point #2 requires extra runs (in search of class-bounding angles) and is expected to lead to modification of the proposed classification. (This would make it major revisions.) Despite the comments on the discussion section, the paper is a valuable addition to knowledge of the influence on flow by bedforms.

**Comments in the pdf**

L6. dunes also occur in a wider range of sediments: silty sediments, sands and gravels (this is also mentioned later in the introduction)

**Removed "sandy"**

L13. no. (1) defines lee-side shape (with the max. angle close to the trough), but no.s (2) and (3) only mention mean lee-side angle (not shape).

**In order to keep the abstract short, we have removed the mention of the maximum angle in (1) but added a sentence that the details of the influence of maximum slope position are discussed in the manuscript.**

**This is now:**
**"We propose a classification with 3 types of dunes: (1) low-angle dunes (mean lee side < 10°), over which there is generally no flow separation and over which only little turbulence is created; (2) intermediate-angle dunes (mean lee side 10-17°) over which an intermittent flow separation is likely over the trough; and (3) high-angle dunes (mean lee side > 17°) over which the flow separates at the brink point and reattaches shortly after the trough, and over which turbulence is high. The maximum lee side slope position has an influence on flow characteristics which depends on dune type. We discuss the implications of the proposed dune classification on the interaction between dune morphology and flow."**

L23. Kleinhans has published many highly relevant papers on high-angle dunes; consider adding one of those.

**We have added a reference to Kleinhans (2004) to the previous sentence.**

L30. And on continental shelves even lower: av. 2 degrees for Netherlands CS (e.g., Damen et al., 2018: Replication data for: Spatially varying environmental properties controlling observed sand wave morphology. 4TU. https://doi.org/10.4121/uuid:0d7e016d-2182-46ea-bc19-cdfda5c20308"). By heart, I think Franzetti et al. 2013 report on giant sand waves (not standard) at a specific site off the coast of Brittany (and generalise into an empirical equation). The dataset of Damen et al. comprises all sand waves on the NCS.

**The angles are not reported in Damen et al (2018) but they are given in the dataset (which I hadn't looked at for a long time, assuming it would be the same data as presented in the paper). I have now added the reference to the dataset.**

L34. very good point

L45. whereas (or start new sentence with However,)

**New sentence started with However**

L46. I agree and interesting point for future research. Small superimposed dunes were observed to change asymmetry direction with the turning of the tide.

L48. please refer here to Figure 1 d and e, respectively, for the max angle position on the slope. This would improve the link between text and figure.

**Changed to "the shapes of river (Figure 1d) and estuarine dunes (Figure 1e) differ"**

excellent introduction

**Thanks!**

Figure 1. "several"" (here and in the figure caption) is slightly vague. Are these the six rivers of Cisneros et al 2020, as mentioned in line 28?
Your point is clearly made with this figure: max angle at 0.4 for rivers and 0.7 for the Weser. Still, the question arose: would you not loose useful information by presenting all rivers in one histogram? E.g., any variation in the position of max lee-slope angle among those rivers.

**The point is made clear that the study from Cisneros et al. (2020) was on six large rivers.**

**There are some differences regarding the position depending on the rivers, but they are not that large. Below a figure with the detail of each river, the number on each subplot shows the median value for this river. The median value for all the rivers is 0.46. In all river cases the maximum slope is predominantly situated in the lower half of the lee slope.**

**I do not think that putting this plot in the paper really brings so much, so I haven't done it. If you feel it is important, I am happy to put it in the Appendix.**

[Figure]

L89. "s"

**Added s at "points"**

L89 ","

**Added a comma after "column"**

L93-96. Since there is no nomenclature section, these last sentences of section 2.1 benefit from adding units in brackets [certainly because dt = 0.0005 minutes is not SI] and symbols with the entities in words. E.g., dt is the time step (minutes? seconds?), g is ..(m s-2) and h is the water depth (m).|u| is a ... (m s-1) [see also Main point #1 in Reviewer's comments.]

**Good point thanks. The timestep in Delft3D is set in min, which is indeed confusing because the equations are of course in SI. I followed your suggestion but also added the timestep in seconds. These sentences now read**

The time step dt was set to 8.33 $10^{-6}$ s (0.0005 min) following a Courant Friedrich Lewy criterion CFL = dt √ (g h) / dx < 10, where dt is the time step (seconds), g is the acceleration due to gravity (m s$^{-2}$) and h is the water depth (m). Since the z-model was used, the following condition also applies: dt ≤ dx / |u| where |u| is a characteristic value of horizontal velocities (m s$^{-1}$) (Deltares, 2014). A uniform background horizontal viscosity of $10^{-3}$ m$^2$ s$^{-1}$ and background vertical eddy viscosity of 0 m$^2$ s$^{-1}$ were set. A k-ε turbulence closure model (Uittenbogaard et al., 1992) was used.

L99. Should be Table 1? Also in the rest of the text.

**Corrected throughout the manuscript**

L117. is there a reason why the 7th? not the 5th, 6th or 8th?

**"in order to characterize equilibrium conditions that are not perturbed by entrance and exit conditions" added**

L123. The introduction and previous studies (Lefebvre, Cisneros) have suggested that intermittend flow separation is a specific flow for many dunes (=intermediate lee angles and shapes; e.g. green crosses in Fig. 3?). Is this not a short coming in this investigation? Please discuss in discussion what the consequences are for the findings of this study. [see also Main point #2 in Reviewer's report]

**As this is a main point, it is not addressed here in the method section but in the discussion**

L138. earlier, notation $m2\ s^{-2}$ was used

**Corrected**

L150. and maximum?

**"The maximum angle stayed as fixed (Figure 2)" added.**

L151. Appendix

**Corrected**

The presentation of these equations looks a bit strange. Perhaps you could present these in a more traditional way. e.g., by introducing a symbol for relative flow separation length and mean lee slope angle; line 159, or already in line 121; which can also be used in the Figures), and so that the equations here can be, e.g. $Lsr = 0.17\ \alpha mean - 0.67$. Etc. [See Main point #1 in Review report]

**It was changed here and at many places in the manuscript following main point #1.**

**"The results from all the simulations (Figure 3) show that the relative flow separation length ($rL_{FSZ}$) and the mean TKE ($TKE_{mean}$) generally increase with increasing mean lee side angle ($\alpha_{mean}$). Both are linearly related to mean lee side angle: $rL_{FSZ} = 0.17\ \alpha_{mean} - 0.67\ (R^2 = 0.70)$ and $TKE_{mean} = 0.00004\ \alpha_{mean} - 0.0009\ (R^2 = 0.87)$."**

L166. i.e. Figure A2 in appendix? If yes, please refer here in the text to the figure.

**Actually, we are not showing all the combination which we tried and do not show any trend. On the other hand, we noticed that Appendix A2 and A3 were not correctly referenced and we corrected this.**

L172. This is not really explained why. You could say ""Based on the flow-bedform correlation results in Figure 3, we identify three categories"" (or something like this).
P.S.: coming back to this after the results and discussion sections: classification is different

afterall. This is confusing (when writing this sentence, you already know the outcome/discussion).

L185. Represents̶

**Corrected**

L196. Add period to al

**Corrected**

L197. Add a dot after al

**Corrected**

L205. or shear layer (i.e., as used in-text)

**"from which the shear layer can be seen" added here and on other figures**

L206. for the straight lee slope and 4 lee shape configurations.

**Added here and on the other figures**

L216 see earlier comment on not being able to model intermittend flow separation, which is expected to be the most occurring for intermediate-angle dunes.

**A sentence is already added here to point this out**

**"This is likely due to the fact that the model can only simulate permanent flow separation whereas intermittent flow separation is most likely to occur over intermediate-angle dunes."**

**And this is further discussed in the discussion**

L219. This seems to be expected to me, whereas the next sentence "As a result" strikes me as a more interesting finding.

**For me this was a surprise. Over high-angle dunes, the flow separates at the brink point, and it is what I expected for the intermediate-angle dunes. Therefore, I left this word here.**

L220. However, Figure 6 shows no flow separation for Config1 (steepest lee slope closest to the crest).

**Yes, as it is stated a couple of lines before ("For example, a flow separation is often absent over dunes where the maximum angle is close to the crest (Figure 6) and / or for sharp profiles"). I added a few words in this sentence as a reminder "As a result, when a flow separation is present, it is longer for maximum angles close to the crest than for maximum angles close to the trough."**

L222. Indeed, is also a very interesting finding.

**To make stand out, we added "moreover" at the beginning of the sentence.**

L224. and extends into the flow (water column), above the dune height.

**Added**

L268. the straight line indicates a constant value over a dune. The slope effect should play a role? (down-slope transport vs. up-slope (gravity))

**Yes, it does play a role, but not that important on the stoss side (because the stoss side angle are so low, around 3°). It also plays a role on the lee side of course, by making it easier for sediment to be entrained down, especially along the steep face.**

**I tried a figure where we show the critical shear stress corrected for slope effect (following Soulsby 1997), and I am not sure it brings so much. There is now so much to see that it gets confusing. The main effect regarding critical bed shear stress is actually from the sediment size. In principle, this should vary along the bedform, which I am not considering here. Honestly, I really feel this should deserve more that my simple analysis here and I am enclined to leave it as it is in this paper.**

[Figure]

**Bed shear stress and bed level for bedforms with a mean lee side of 5°, 15° and 25° and with a straight lee side (left panel), or a maximum angle of 30° close to the crest (config1, middle panel) or close to the trough (config4, right panel). The dotted lines on the upper plots show the critical shear stress for bed load transport (adjusted for slope effect)**

**I have added a few sentences at the beginning of section 4.4. to highlight that this is a simple analysis.**

"Here, it should be noted that the critical shear was calculated using a single sediment size and without bed slope effect (following the procedure described by Soulsby (1997)), and of course, it does not take into account the complex feedback between sediment transport and morphology variation (i.e. no sediment is being moved). It is therefore a simple indication for the potential to put sediment in motion along the dune."

L274. among?

**Changed.**

L274. see main comment #2 in review report

L274. 's' **added**

L278. add 'not'? (not specifying?)

**Changed to "did not specify"**

L278-279. This is the same as you suggest. Perhaps not explicitly naming intermediate, but that is definitely implied (between low and high angle).

**That's true. Therefore I have changed the word "differs" in the previous sentence into "adds precision".**

L282. implicitly? that should be enough, see previous comment.

**Corrected**

L284. I do not entirely agree. See main comment #2 in review report

Figure 9. intermediate angle. independent of max slope position - also independent of max slope position

**First one added, the second one not added**

"Figure 9: in this summarising figure (where text is used to further explain): do you really need the two configurations? (hard to see the differences anyway). The conceptual value of Figure 9 would increase if it would schematise (not trying to mimic figs 4-7) the main, most discriminating points. For example, low angle dunes: long shear layer; wakes within the height of dunes; lengths of shear and wakes dependent on lee shape.
Fig 9: For the (variation in) lengths of wakes, e.g. use a red (or yellow) dot to indicate the position of steepest slope. For longest wake high-angle dunes, indicate with a red dot: largest when steep slope at top of lee side close to the crest. "

**I played around a bit with the figure. I left the two configurations for low and high-angle dunes because the position of the maximum slope has an influence (and I indicated the position of the maximum slope with a red dot as suggested) but I removed it for the intermediate-angle dune**

L302. Lightly -> **slightly**

L303. see main comment #2

L304. see main comment #2

L319. Wake **added**

L340. Galeazzi et al 2018 as well.

**Added**

L369. low-angle dunes + better to place this earlier in the sentence.

**Sentences changed to**

**"For low-angle (mean lee side <10°) dunes, the extended area of critical bed shear stresses near the crest in case the maximum angle is near the crest (config1) implies that there is higher ability of sediment to be transported from the dune crest compared to the configuration where the maximum angle is near the trough (config4)."**

L379-384. This is a repetition of the results. Why not relate to the critical shear stress for incipient motion, as indicated in Fig. 8 (see earlier comment that slope effect should be taken into account for ciritical shear stress?). Also, interesting in discussing the consequences for sediment transport in this section (4.4) would be:

- does the detachment of the shear layer in higher angle cases lead to more suspended sediment i.s.o. bedload?

- the line of critical shear stress (Fig 8) implies that sediment will only be mobilised at the crests of dunes. However, empirical studies show that large parts of stoss sides are being eroded. Is this a consequence of the choice of initial dune shape in the simulation runs?

**This paragraph was largely rewritten and these points considered.**

L442. see main comment #1. Better to write as equation

TKEmax = 12.4581 * TKEmean - 0.0079"

**Done**

---

## Referee Report (RR1)

Reply Reviewer #2 to author's comments (AC1) and revised manuscript egushere-2023-211

I have read the author's comments (revisions) in the interactive discussion and am very happy to see that the authors added simulations to investigate the boundaries between classes of dunes to strengthen their dune classification (Appendix 4 clarifies this; strong addition to the paper). And also, happy to see that they did a few tests to answer the questions that arose from the first manuscript. The revised manuscript now reads as a consistent paper. The extensive results in excellent figures with improved discussion all valuable contribute to the knowledge of the impact of dune lee-slope shape on flow (special thanks for rewriting lines 409-416 in the revised manuscript). A great paper to publish!

In reply to their comments, I very much like the figure with the slope-dependent critical shear stress for incipient motion. Because the effect is only small and the figure more complicated to read, the authors decided to not use this figure and added text to the discussion (paragraph 4.4; which is good); perhaps it is an idea to mention this (still) in the caption of Figure 8, e.g. add at the end "(not adjusted for slope effect, see section 4.4)"? This is just a suggestion, = up to the authors.

With the added paragraph on three-dimensionality of dunes, the modelling work of Nathaniel Bristow (over barchans) is interesting as well. (Just pointing out; no need to add to the manuscript, because that would need explanation barchans vs river dunes etc. and there are ample references in the text as it is.)

All new in-text citations all seem to have been added to the Ref List (merely quick scanned).

A few minor text corrections in revised manuscript:
Line 246: High-angle dunes.... represent (delete 's' in represents)
Line 389: add a period at the end of this last sentence (By the way for all figure captions, a period could be added at the end.)
Line 394: Furthermore, the dune shape investigate is (shape=singular); or, preferably: dune shapes investigated are (=plural)

Looking forward to seeing this as a publication.

---

## Author Response (AR2)

Reply Reviewer #2 to author's comments (AC1) and revised manuscript egushere-2023-211

I have read the author's comments (revisions) in the interactive discussion and am very happy to see that the authors added simulations to investigate the boundaries between classes of dunes to strengthen their dune classification (Appendix 4 clarifies this; strong addition to the paper). And also, happy to see that they did a few tests to answer the questions that arose from the first manuscript. The revised manuscript now reads as a consistent paper. The extensive results in excellent figures with improved discussion all valuable contribute to the knowledge of the impact of dune lee-slope shape on flow (special thanks for rewriting lines 409-416 in the revised manuscript). A great paper to publish!

In reply to their comments, I very much like the figure with the slope-dependent critical shear stress for incipient motion. Because the effect is only small and the figure more complicated to read, the authors decided to not use this figure and added text to the discussion (paragraph 4.4; which is good); perhaps it is an idea to mention this (still) in the caption of Figure 8, e.g. add at the end "(not adjusted for slope effect, see section 4.4)"? This is just a suggestion, = up to the authors.
With the added paragraph on three-dimensionality of dunes, the modelling work of Nathaniel Bristow (over barchans) is interesting as well. (Just pointing out; no need to add to the manuscript, because that would need explanation barchans vs river dunes etc. and there are ample references in the text as it is.)

All new in-text citations all seem to have been added to the Ref List (merely quick scanned).
A few minor text corrections in revised manuscript:
Line 246: High-angle dunes.... represent (delete 's' in represents)
Line 389: add a period at the end of this last sentence (By the way for all figure captions, a period could be added at the end.)
Line 394: Furthermore, the dune shape investigate is (shape=singular); or, preferably: dune shapes investigated are (=plural)

Looking forward to seeing this as a publication

Dear reviewer,

Thank you very much for reading our revised manuscript. We are very happy that our revisions are appreciated. We have corrected the three small corrections that you suggested, as well as added "not adjusted for slope effect" in the caption of Figure 8 as suggested.

Dear Alice and Julia,
Thank you for all your hard work with the manuscript revisions. I am delighted to inform you that the paper has now been accepted for final publication subject to 'technical corrections'. This means there are 2-3 minor changes as identified by one of the reviewers that can be carried out without any further need for review etc.. If you could carry these out then the paper can proceed to final publication.
All the best,
Tom Coulthard

three minor technical corrections required as identified by the reviewer in their report,

Dear editors,
We are very happy with the handling of our manuscript and pleased that it can be published by Earth Surface Dynamics as an open access paper.

Alice & Julia